# Rethinking Preference Alignment for Diffusion Models with Classifier-Free Guidance

## Abstract

Aligning large-scale text-to-image diffusion models with nuanced human preferences remains challenging. While direct preference optimization (DPO) is simple and effective, large-scale finetuning often shows a generalization gap. We take inspiration from test-time guidance and cast preference alignment as classifier-free guidance (CFG): a finetuned preference model acts as an external control signal during sampling. Building on this view, we propose a simple method that improves alignment without retraining the base model. To further enhance generalization, we decouple preference learning into two modules trained on positive and negative data, respectively, and form a *contrastive guidance* vector at inference by subtracting their predictions (positive minus negative), scaled by a user-chosen strength and added to the base prediction at each step. This yields a sharper and controllable alignment signal. We evaluate on Stable Diffusion 1.5 and Stable Diffusion XL with Pick-a-Pic v2 and HPDv3, showing consistent quantitative and qualitative gains.

## 1 Introduction

Diffusion models (Ho et al., 2020; Song & Ermon, 2019; Song et al., 2021) are one of the most prevalent generative models for high-fidelity text-to-image (T2I) synthesis (Podell et al., 2023; Saharia et al., 2022). These models are typically trained from Internet-scale datasets which, due to the tremendous scale, are not carefully curated. A diffusion model pretrained on these datasets therefore deviate from what humans (in the majority voting sense) truly prefer in aspects such as aesthetic and instruction following (Kirstain et al., 2023).

The same issue is well studied in the field of large language model (LLM), in which naïvely pretrained LLMs without any post-training steps do not follow instructions and are not able to chat naturally with human (Ouyang et al., 2022) . Typical approaches to align LLMs with human preference for LLMs include 1) reinforcement learning from human feedback (RLHF) (Ouyang et al., 2022), which demands a reward model pretrained on a preference dataset and requires careful hyperparameter tuning, and 2) direct preference optimization (DPO) (Rafailov et al., 2023b), the simpler alternative that bypasses reward modeling by essentially treating the alignment problem as a binary classification problem on positive-negative preference pairs. This simple solution of DPO can be easily adapted for aligning diffusion models with human preference such as the method of Diffusion-DPO (Wallace et al., 2024) and has been widely used in applications other than T2I synthesis (Wang et al., 2023; Blattmann et al., 2023; Wu et al., 2023a; Khachatryan et al., 2023). Nevertheless, DPO is generally considered less robust compared to RLHF: it is prone to overfitting, may produce non-smooth predictions on out-of-distribution text prompts, and even exhibit catastrophic forgetting behaviors (Lin et al., 2024). While one may include in either the whole pretraining dataset or just the prompt set to regularize models, access to these pretraining sets is typically infeasible for large-scale models.

We conduct a toy 2D experiment (Fig. 1) to illustrate how DPO fails. Specifically, we consider a toy 8-Gaussians dataset composed of 2D points sampled from 8 Gaussian balls, for which we label half of the Gaussian balls as positive samples and the rest of them as negative ones. With this dataset, we train an simple diffusion model parameterized by a 3-layer-MLP to perform Diffusion-DPO finetuning with randomly sampled preference pairs. Even on this toy dataset, DPO-tuned models

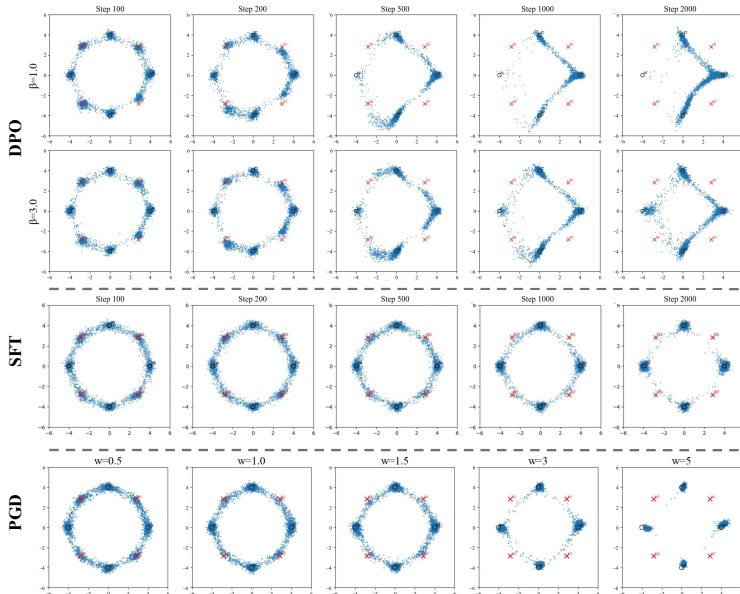

Figure 1: Toy 2D experiment on DPO (top) , SFT (Middle) and our proposed PGD (below) to demonstrate the overfitting issue in DPO training. Black circles indicate positive sample clusters and red crosses indicate negative sample clusters. $w$ is the CFG scale and $\beta$ is the DPO scale parameter.

deviate from ideal finetuned distributions, and, if trained for too long, easily suffers from overfitting and mode collapse.

We instead take inspiration from inference-time techniques for diffusion model adaptation. Specifically, we observe that classifier free guidance (CFG) (Dhariwal & Nichol, 2021), the standard approach for sampling from conditional diffusion models by linearly combining between unconditional and conditional predictions, can be viewed as tempering the potentially overfitted conditional model with the more generalizable unconditional prior, with the shape of the posterior free to control during test time. Since the posterior distribution obtained through CFG typically exhibits strong performance, and the alignment objective from the control as inference perspective (Levine, 2018) is likewise to obtain a posterior distribution (Rafailov et al., 2023a), we are led to ask: *can CFG be adopted to address the diffusion alignment problem?*

Motivated by this question, we view a finetuned diffusion model as the diffusion model conditioned on a virtual control signal from the preference dataset, while the base model serves as the unconditional model. From this perspective, sampling from the aligned diffusion model naturally becomes a CFG-style inference process, which gives rise to our first method, Preference-Guided Diffusion (PGD). With the CFG scale to amplify the difference between the control signal and the prior during test time, the control signal does not have to be a fully-finetuned model but one finetuned only with a few iterations, thus effectively preventing overfitting. Adopting this CFG perspective further suggests that finetuning should resemble conditional diffusion training, which does not rely on positive–negative pairs but instead uses the standard diffusion loss. To implement this idea, we finetune two models independently, one that generates positive samples and another that generates negative samples, and combine them at inference through CFG-style composition. We refer to this variant as contrastive Preference-Guided Diffusion (cPGD). In experiments, PGD and cPGD consistently outperform vanilla Diffusion-DPO. Notably, both methods achieve Pareto improvements, simultaneously yielding higher reward, lower FID, and greater diversity in the generated samples. Moreover, the approach in principle produces a transferable plug-and-play module that, once trained on a base diffusion model, can be reused to align others.

In summary, our contributions are

- We propose to alleviate the generalization issue in Diffusion-DPO by treating diffusion model alignment as a special case of CFG-style inference.
- We introduce Preference-Guided Diffusion (PGD), which aligns the generated distribution with human preference through CFG-style guidance at inference time.

- We extend this view by considering finetuning as conditional diffusion training and propose contrastive PGD (cPGD).

- We empirically demonstrate that both variants achieve Pareto improvements over the Diffusion-DPO baseline.

## 2 RELATED WORK

**Alignment with human preference.** Preference optimization has become central to aligning large generative models with human expectations. In large language models, reinforcement learning from human feedback (RLHF) (Ouyang et al., 2022) is the dominant framework, relying on a reward model trained from pairwise human preferences (Christiano et al., 2017; Stiennon et al., 2020). While effective, RLHF requires careful hyperparameter tuning in both the reward model and reinforcement learning stages. In contrast, direct preference optimization (DPO) (Rafailov et al., 2023b), its diffusion-specific extension Diffusion-DPO, and several related alternatives (Azizzadenesheli et al., 2023; Xu et al., 2024a; Lin et al., 2024) offer a simpler approach: directly finetuning the model with a logistic regression objective on preference pairs, thereby eliminating the need for an explicit reward model. However, DPO methods are often less competitive than RLHF (Ouyang et al., 2022), a limitation also observed in recent adaptations of preference optimization to text-to-image diffusion models (Black et al., 2023; Lee et al., 2023; Black et al., 2024; Fan et al., 2023; Xu et al., 2024b; Clark et al., 2024; Prabhudesai et al., 2023; Wallace et al., 2024; Li et al., 2024; Yang et al., 2024; Zhu et al., 2025). Building on this line of work, we propose a Diffusion-DPO variant that reformulates preference alignment as inference-time guidance to improve generalization.

**Guidance in diffusion models.** Controlling diffusion models can be broadly categorized into *fine-tuning approaches* and *inference-time guidance approaches*. Fine-tuning methods adapt model parameters to inject conditioning signals or domain knowledge. Representative examples include DreamBooth (Ruiz et al., 2023), which personalizes text-to-image models with subject-specific data, and other adapter- or LoRA-style techniques (Hu et al., 2021; Gal et al., 2022). While effective, such methods require additional training and may incur overfitting or catastrophic forgetting when data is limited. In contrast, inference-time guidance requires no additional training and modifies the sampling process to incorporate conditioning. Classifier guidance (Dhariwal & Nichol, 2021) uses the gradient of an external classifier, but can lead to distributional shifts. Classifier-free guidance (CFG) (Ho & Salimans, 2022) avoids this by training with randomly dropped conditions and linearly combining between unconditional and conditional predictions at inference, and has since become the de facto standard for controllable text-to-image generation. Numerous extensions build on this principle, e.g., language-model-based steering (Nichol et al., 2021), attention-based semantic guidance (Chefer et al., 2023), or plug-and-play conditioning modules (Liu et al., 2023). Our work draws direct inspiration from inference-time guidance. Instead of conditioning on textual prompts or class labels, we extend the CFG principle to *preference alignment*, treating human preference as a conditioning signal that can be injected at inference to steer generation toward preferred outputs.

## 3 PRELIMINARIES

### 3.1 DIFFUSION MODELS

Diffusion models are a category of generative models that generate samples by sequentially denoising noisy samples. Specifically, a diffusion model defines a noising (forward) process

$$q(\mathbf{x}_t \mid \mathbf{x}_{t-1}) = \mathcal{N}\big(\sqrt{\alpha_t}\,\mathbf{x}_{t-1},\,(1-\alpha_t)\mathbf{I}\big). \tag{1}$$

where $\{\beta_t\}_{t=1}^T$ and $\alpha_t = 1 - \beta_t$, $\bar{\alpha}_t = \prod_{s=1}^t \alpha_s$ are the noise schedule, typically set such that $q(x_T|x_0) \approx \mathcal{N}(0, I)$. Sampling from this diffusion model is through the denoising (reverse) process:

$$p_\theta(\mathbf{x}_{t-1} \mid \mathbf{x}_t) = \mathcal{N}\big(\boldsymbol{\mu}_\theta(\mathbf{x}_t, t),\, \sigma_t^2 \mathbf{I}\big), \qquad \boldsymbol{\mu}_\theta(\mathbf{x}_t, t) = \frac{1}{\sqrt{\alpha_t}}\left(\mathbf{x}_t - \frac{\beta_t}{\sqrt{1-\bar{\alpha}_t}}\,\epsilon_\theta(\mathbf{x}_t, t)\right), \quad (2)$$

with $\sigma_t^2$ set to the posterior variance $\tilde{\beta}_t = \frac{1-\bar{\alpha}_{t-1}}{1-\bar{\alpha}_t}\beta_t$.

Training a diffusion model amounts to simply minimize the diffusion loss

$$\mathcal{L}_{\text{DDPM}}(\theta) = \mathbb{E}_{t,\mathbf{x}_0,\epsilon}\Big[ w(t)\big\|\epsilon - \epsilon_\theta\big(\sqrt{\bar{\alpha}_t}\,\mathbf{x}_0 + \sqrt{1-\bar{\alpha}_t}\,\epsilon,\, t\big)\big\|_2^2\Big] \tag{3}$$

where $w(t)$ is a weighting scalar with one of the common choices being $w(t) = 1$. Such as objective is equivalent to matching the model output $\epsilon_\theta(x, t)$ with the ground-truth score function $\nabla \log p_t(x)$ and we therefore use $\nabla \log \pi(x, t; \theta)$ interchangeably with $\epsilon_\theta(x, t)$.

## 3.2 DIRECT PREFERENCE OPTIMIZATION

Given a preference dataset $\mathcal{D} = \{(x_+^{(i)}, x_-^{(i)}, c^{(i)})\}_{i=1}^N$, where $x_+$ and $x_-$ denote the preferred and dispreferred samples conditioned on a prompt $c$, direct preference optimization (DPO) (Rafailov et al., 2023a) performs logistic regression on the relative log-odds:

$$L_{\text{DPO}} = - \mathbb{E}_{(x_+, x_-, c) \sim \mathcal{D}} \left[ \log \sigma \Big( \beta \log \frac{\pi_\theta(x_+|c)}{\pi_{\text{ref}}(x_+|c)} - \beta \log \frac{\pi_\theta(x_-|c)}{\pi_{\text{ref}}(x_-|c)} \Big) \right], \tag{4}$$

where $\sigma(\cdot)$ is the sigmoid function and $\beta > 0$ is an inverse-temperature. Equivalently, DPO can be viewed as maximum likelihood under an implicit reward model (with normalization constant $Z$) (Rafailov et al., 2023b): $r(x, c) = \beta \log \frac{\pi^*(x|c)}{\pi_{\text{ref}}(x|c)} + \log Z$, together with the Bradley–Terry (BT) preference model (Bradley & Terry, 1952) that, under the optimal policy $\pi^*$, yields

$$p(x_+ \succ x_- \mid c) = \sigma \Big( \beta \log \frac{\pi^*(x_+|c)}{\pi_{\text{ref}}(x_+|c)} - \beta \log \frac{\pi^*(x_-|c)}{\pi_{\text{ref}}(x_-|c)} \Big). \tag{5}$$

**DPO for diffusion models.** A naïve application of DPO to diffusion models would treat $x_+$ and $x_-$ as entire reverse-diffusion trajectories from $t=T$ to $t=0$. This is intractable because computing trajectory-level likelihood ratios requires integrating over all intermediate noise steps. Diffusion-DPO (Wallace et al., 2024) circumvents this by applying Jensen's inequality to obtain an upper bound on the trajectory loss. Decomposing the joint log-likelihood into a sum of per-step transition log-likelihoods leads to a simple transition-wise objective:

$$L = - \mathbb{E}_{\substack{(x_0^+, x_0^-, c) \sim \mathcal{D} \\ t \sim U\{1,...,T\}}} \left[ \log \sigma \Big( \beta \log \frac{\pi_\theta(x_+^{(t-1)}|x_+^{(t)}, c)}{\pi_{\text{ref}}(x_+^{(t-1)}|x_+^{(t)}, c)} - \beta \log \frac{\pi_\theta(x_-^{(t-1)}|x_-^{(t)}, c)}{\pi_{\text{ref}}(x_-^{(t-1)}|x_-^{(t)}, c)} \Big) \right], \tag{6}$$

where $\pi_\theta(x_{t-1} \mid x_t, c)$ denotes the one-step reverse-diffusion transition under $\pi_\theta$.

Approximating the transition log-likelihoods with standard diffusion losses then yields:

$$L_{\text{Diff-DPO}}(\theta) = - \mathbb{E}_{\substack{(x_0^+, x_0^-, c) \sim \mathcal{D} \\ t \sim U\{1,...,T\}}} \left[ \log \sigma \Big( - \beta T \omega(\lambda_t) \left( \|\epsilon^+ - \epsilon_\theta(x_+^{(t)}, t, c)\|_2^2 - \|\epsilon^+ - \epsilon_{\text{ref}}(x_+^{(t)}, t, c)\|_2^2 \right. \right.$$

$$\left. \left. - \|\epsilon^- - \epsilon_\theta(x_-^{(t)}, t, c)\|_2^2 + \|\epsilon^- - \epsilon_{\text{ref}}(x_-^{(t)}, t, c)\|_2^2 \right) \Big) \right]. \tag{7}$$

where $\epsilon^+$ and $\epsilon^-$ are the Gaussian noises used to form $x_+^{(t)}$ and $x_-^{(t)}$ from $x_0^+$ and $x_0^-$, $\epsilon_\theta$ and $\epsilon_{\text{ref}}$ are the model and reference noise predictors, $\omega(\lambda_t)$ is the usual weighting as a function of noise level $\lambda_t$, and $T$ is the number of diffusion steps.

## 3.3 CONDITIONAL GENERATION AND CLASSIFIER-FREE GUIDANCE

With a conditional diffusion model trained on the dataset $\{(x_i, c_i)\}_{i=1}^N$, the inference process typically adopts classifier-free guidance (CFG) (Ho & Salimans, 2022) that samples instead with the composed score estimate:

$$\hat{\epsilon}(\mathbf{x}_t, t, \mathbf{c}) = \epsilon_u + w \cdot (\epsilon_c - \epsilon_u), \tag{8}$$

where $\epsilon_u = \epsilon_\theta(\mathbf{x}_t, t, \varnothing), \epsilon_c = \epsilon_\theta(\mathbf{x}_t, t, \mathbf{c})$ are the unconditional and conditional score estimate (respectively), $w$ is a positive guidance weight that is usually greater than 1, and $\varnothing$ is the null condition. In practice, $\epsilon_u$ is trained by setting the embedding of the condition input to zero. The CFG inference process approximately generates samples from the posterier distribution $p(x)p^w(c|x)$ with $p(x)$ being the prior distribution, or equivalently in log-likelihood, $\log p(x) + w \log p(c|x)$.

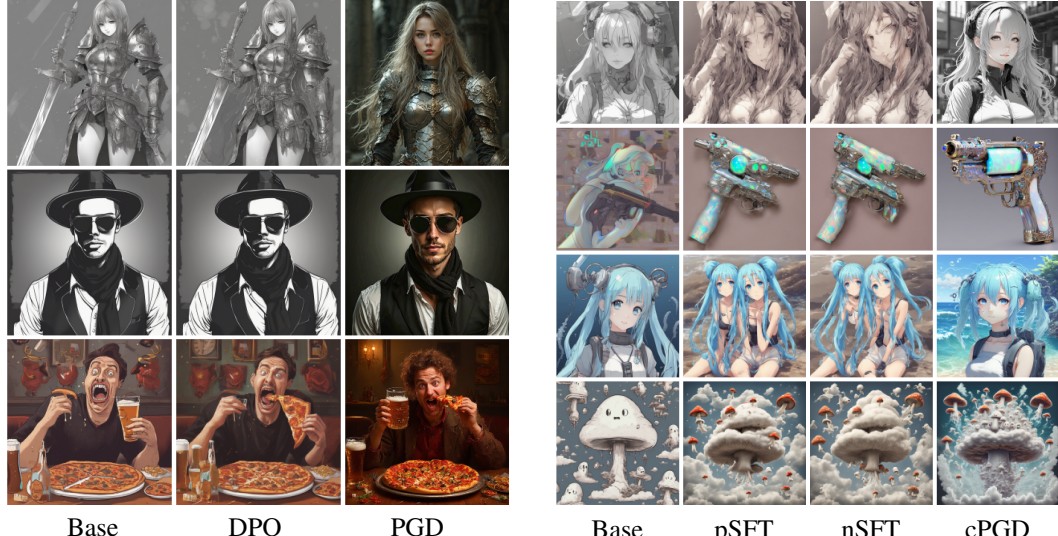

Figure 2: Comparison of base, DPO, and PGD: PGD retains base fidelity while leveraging DPO-learned preferences.

Figure 3: Illustration of cPGD. pSFT and nSFT denote inference with the model finetuned on positive and negative samples, respectively.

# 4 METHOD

## 4.1 PREFERENCE-GUIDED INFERENCE

Let $\pi_{\text{ref}}$ be a reference policy and $\pi_{\text{DPO}}$ be a DPO-tuned policy. By treating the DPO-tuned policy as $\pi(x|\mathcal{D})$ and the reference policy as $\pi(x|\varnothing)$ as in CFG, we immediately obtain the CFG-style score function for inference, for which we term preference-guided diffusion (PGD):

$$\nabla \log \pi_{\text{PGD}}(x) = \nabla \log \pi_{\text{ref}}(x) + w\Big(\nabla \log \pi_{\text{DPO}}(x) - \nabla \log \pi_{\text{ref}}(x)\Big), \qquad (9)$$

where the guidance weight $w$ determines the trade-off between our confidence on the reward and other metrics such as prior preservation and sample diversity. Since $\pi_{\text{ref}}$ can be understood as some prior pretrained on unlabeled datasets, once we have trained $\nabla \log \pi_{\text{DPO}}(x)$, we are able to virtually align any other base model $\pi'_{\text{ref}}(x)$ by simply replacing $\pi_{\text{ref}}(x)$ with it.

## 4.2 CONTRASTIVE PGD AS DYNAMICALLY-REWEIGHTED GUIDANCE

The connection between CFG and diffusion model alignment prompts us to think whether finetuning should also be done in a way similar to conditional diffusion model training, which directly encourage the negative score functions to point towards the data points. However, our preference dataset contains both positive samples and negative ones. These negative samples act as "repelling" forces that pushes the negative score function away from them. Inspired by that much of alignment can be turned into an inference-time manner, we propose to postpone this "repelling" behavior to inference-time as well. Specifically, we finetune another copy of the base model so that it generates negative samples. Formally speaking, with $\mathcal{D}_+$ representing the set of positive samples and $\mathcal{D}_-$ the set of negative ones, we independently finetune two models (with parameters $\theta_+$ and $\theta_-$, respectively) with diffusion losses:

$$L_{\text{pos}}(\theta_+) = \mathbb{E}_{\epsilon \sim \mathcal{N}(0,I), t \sim \text{Uniform}(0,1), x_0 \sim \mathcal{D}_+} \left\| \epsilon - \epsilon\big(\sqrt{\bar{\alpha}_t}\, x_0 + \sqrt{1 - \bar{\alpha}_t}, \epsilon, t; \theta_+\big) \right\|^2 \qquad (10)$$

$$L_{\text{neg}}(\theta_-) = \mathbb{E}_{\epsilon \sim \mathcal{N}(0,I), t \sim \text{Uniform}(0,1), x_0 \sim \mathcal{D}_-} \left\| \epsilon - \epsilon\big(\sqrt{\bar{\alpha}_t}\, x_0 + \sqrt{1 - \bar{\alpha}_t}, \epsilon, t; \theta_-\big) \right\|^2 \qquad (11)$$

Intuitively, the difference between two models characterizes the implicit reward model. Therefore we may write the residual parameterization $\nabla \log \pi_{\text{finetuned}}(x,t) = \nabla \log \pi(x,t;\theta_+) - \nabla \log \pi(x,t;\theta_-) + \nabla \log \pi_{\text{ref}}(x,t)$. It follows that the resulting PGD formulation, to which we

refer with contrastive PGD (cPGD), is

$$\nabla \log \pi_{\text{PGD}}(x, t) = \nabla \log \pi_{\text{ref}}(x, t) + w \Big( \nabla \log \pi(x, t; \theta_+) - \nabla \log \pi(x, t; \theta_-) \Big). \qquad (12)$$

**Alternative perspective of cPGD.** While it may seem a bit arbitrary to replace the DPO loss on the preference dataset with two diffusion losses on positive-only and negative-only datasets respectively, cPGD essentially performs dynamic reweighting of DPO loss gradients. For simplicity, let's consider the general DPO case (without Diffusion-DPO approximations). If we plug the residual parametrization of the finetuned model into the DPO loss gradient, we observe (with $\theta = (\theta_+, \theta_-)$):

$$\nabla_\theta L_{\text{DPO}} = - \mathop{\mathbb{E}}_{(x_+, x_-) \sim \mathcal{D}} \Big[ \beta \sigma \Big( \log \pi(x; \theta_-) - \log \pi(x; \theta_+) \Big)$$
$$\cdot \Big( \nabla_{\theta_+} \log \pi(x; \theta_+) - \nabla_{\theta_-} \log \pi(x; \theta_-) \Big) \Big]. \qquad (13)$$

Suppose that, for each sample pair $(x_+, x_-)$, we dynamically reweight the loss function by $1 / \Big[ \beta \sigma \Big( \log \pi(x; \theta_-) - \log \pi(x; \theta_+) \Big) \Big]$. The resulting dynamically-reweighted loss is

$$\nabla_\theta L_{\text{reweight}} = - \mathop{\mathbb{E}}_{(x_+, x_-) \sim \mathcal{D}} \Big[ \nabla_{\theta_+} \log \pi(x; \theta_+) - \nabla_{\theta_-} \log \pi(x; \theta_-) \Big]$$
$$= \mathop{\mathbb{E}}_{x_- \sim \mathcal{D}_-} \Big[ \nabla_{\theta_-} \log \pi(x; \theta_-) \Big] - \mathop{\mathbb{E}}_{x_+ \sim \mathcal{D}_+} \Big[ \nabla_{\theta_+} \log \pi(x; \theta_+) \Big] \qquad (14)$$

which is exactly the gradient of the cPGD training objectives once we take into consideration the fact that $\log \pi$ is parameterized by a diffusion model. Wu et al. (2025) show that such reweighting can be seen as an interpolation between supervised finetuning gradients and vanilla policy gradients, and that it can be helpful to alleviate the overfitting issue due to the small scale of finetuning datasets.

## 5 EXPERIMENTS

### 5.1 EXPERIMENTAL SETUP

**Training datasets.** We consider consider two datasets: 1) Pick-a-Pic v2 (Kirstain et al., 2023), which consists of approximately 900,000 image preference pairs derived from 58,000 unique prompts, and 2) HPDv3 (Ma et al., 2025), which comprises 1.08M text-image pairs and 1.17M annotated pairwise data. Our main experiments are done on Pick-a-Pic v2, while for ablation, we create a high-image-quality subset of HPDv3 besides the full dataset of HPDv3.

**Test prompts.** We consider the following prompt datasets for testing: the test split of Pick-a-Pic v2 (424 prompts), the HPDv2 test set (Wu et al., 2023b) (400 prompts), and the Parti-Prompts benchmark (Yu et al., 2022) (1,632 prompts).

**Baselines.** We benchmark our approaches against the following baselines: (i) SFT-Pref, a supervised fine-tuning baseline using only the preferred images; (ii) Diffusion-DPO (Wallace et al., 2024), an adaptation of the DPO method to diffusion models; (iii) Diffusion-KTO (Li et al., 2024), a variant that incorporates a Kullback–Leibler trade-off to Diffusion-DPO for unlocking the potential of leveraging readily available per-image binary signals; (iv) MaPO (Hong et al., 2024), which refines preference optimization with margin-based pairwise consistency; (v) Diffusion-NPO (Wang et al., 2025), which explicitly models negative preferences to strengthen classifier-free guidance. Additionally, we consider SPO (Liang et al., 2024), a hybrid method that trains auxiliary reward models in an online fashion during DPO finetuning; due to its online nature, we exclude SPO from the direct comparison between offline DPO variants.

**Reward models.** We evaluate generated images with these reward models: PickScore (PS) (Kirstain et al., 2023), HPSv2 (Wu et al., 2023b), HPSv3 (Ma et al., 2025), ImageReward (IR) (Xu et al., 2024b), CLIP Score (Radford et al., 2021), and Aesthetics Score (Aes) (Schuhmann, 2022).

**Metrics.** Besides the absolute reward values, we compute win rates for different methods, which is the percentage of instances where the finetuned model outperforms the base model. Since win rates are considerably more robust than absolute reward values (Wu et al., 2023b; Kirstain et al., 2023), we use win rate as our primary metric. In addition, we compute FID score and diversity score to measure the extent of prior preservation and sample diversity, respectively. Sample diversity is

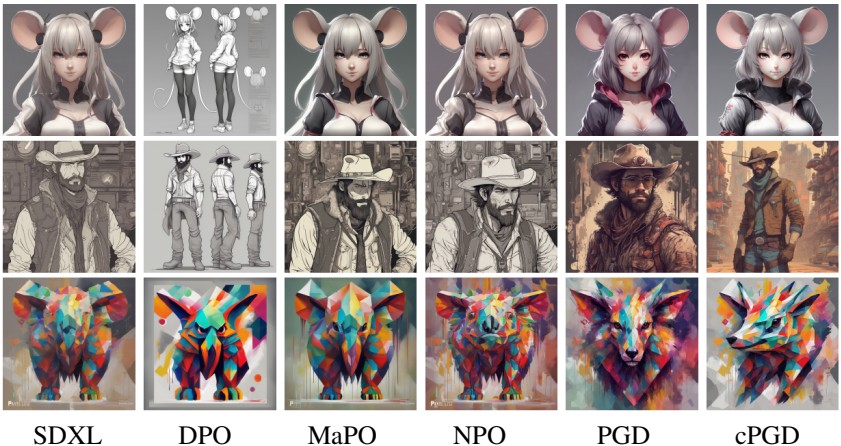

SDXL     DPO     MaPO     NPO     PGD     cPGD

Figure 4: Comparison of preference-optimization methods on SDXL. Columns show outputs from the base model (SDXL), DPO, MaPO, NPO, PGD, and cPGD. PGD and cPGD achieves the highest rewards and is the most effective in aligning with human preference implied in the Pick-a-Pic v2 dataset.

computed by measuring the average pairwise distances between CLIP embeddings Radford et al. (2021) of generated samples (details in Appendix A.2).

**Implement Details.** We experiment with two base models: Stable Diffusion v1.5 (SD1.5) (Rombach et al., 2022) and Stable Diffusion XL base (SDXL) (Podell et al., 2023). An effective batch size of 2048 image pairs is used for all experiments. Following common practices, we set for each model a base learning rate and scale it linearly with the batch size. For SD1.5, we use AdamW optimizer with a base learning rate of 3e-8; for the larger SDXL model, we employ Adafactor optimizer with a base learning rate of 5e-9. Following Wallace et al. (2024), $\beta$ is set to 3000 and 5000 for SD1.5 and SDXL, respectively. It is worth noting, however, that our effective learning rate is smaller than that of Wallace et al. (2024). In their setting, the effective learning rate is 2.048e-5, whereas ours is only 9.6e-7, an order of magnitude smaller. For PGD, we use the finetuned model with training 2000 steps and for cPGD, we use models trained with 500 steps.

## 5.2 RESULTS

| Base Model | Inference Strategy | Pick-a-Pic v2 test (424 prompts) | | | | | | Parti-Prompts (1632 prompts) | | | | | | Avg. ↑ |
|---|---|---|---|---|---|---|---|---|---|---|---|---|---|---|
| | | PS ↑ | HPSv2 ↑ | HPSv3 ↑ | Aes ↑ | CLIP ↑ | IR ↑ | PS ↑ | HPSv2 ↑ | HPSv3 ↑ | Aes ↑ | CLIP ↑ | IR ↑ | |
| SDXL | – | 50.0 | 50.0 | 50.0 | 50.0 | 50.0 | 50.0 | 50.0 | 50.0 | 50.0 | 50.0 | 50.0 | 50.0 | 50.0 |
| | NPO | 58.7 | 59.2 | 69.1 | **52.1** | 37.5 | 53.5 | 55.0 | 56.4 | 62.1 | 55.1 | 39.0 | 51.5 | 54.1 |
| | PGD | 78.8 | 79.0 | 73.6 | 51.9 | 63.7 | 69.3 | **78.7** | **78.4** | 78.2 | **68.8** | 54.0 | 69.1 | 70.3 |
| | cPGD | **80.0** | **80.2** | **77.1** | 50.9 | **64.9** | **69.8** | 75.8 | **80.3** | 77.8 | 57.2 | **62.0** | **74.0** | **70.8** |
| DPO-SDXL | – | 71.7 | 77.6 | 67.9 | 53.3 | 61.6 | 65.8 | 64.0 | 70.3 | 64.0 | 57.5 | 58.5 | 69.7 | 65.2 |
| | NPO | 76.9 | 81.8 | 81.4 | 53.8 | 57.8 | 70.8 | 70.6 | 78.4 | 77.5 | 63.4 | 56.3 | 70.8 | 70.0 |
| | PGD | **83.3** | **85.4** | **85.6** | 59.7 | **62.3** | **73.6** | **80.8** | **83.9** | **81.7** | **67.6** | 57.5 | **76.1** | **74.8** |
| | cPGD | 80.9 | 77.6 | 84.7 | **63.9** | 58.7 | 64.9 | 73.9 | 79.2 | 72.8 | 59.4 | **64.1** | 74.4 | 71.2 |
| MaPO-SDXL | – | 55.9 | 65.3 | 61.8 | 68.2 | 50.2 | 68.2 | 52.0 | 72.4 | 58.5 | 48.2 | 65.0 | 60.8 | |
| | PGD | 80.4 | 81.6 | 81.6 | **75.7** | 51.4 | 72.2 | 78.9 | 77.8 | 79.6 | **77.8** | 53.6 | 72.7 | **73.6** |
| | cPGD | 77.4 | 78.8 | 72.9 | 69.1 | 59.4 | 72.4 | 72.5 | **81.1** | 76.9 | 70.1 | **58.2** | **74.4** | 71.9 |
| SPO-SDXL | – | 89.4 | 83.0 | 96.0 | 81.8 | 33.3 | 78.8 | 87.8 | 85.5 | 92.2 | 88.1 | 31.7 | 74.9 | 76.9 |
| | PGD | 92.2 | 86.1 | 96.5 | 82.1 | 42.5 | 81.4 | 91.4 | 87.7 | 93.9 | 88.4 | 48.0 | 77.3 | 80.6 |
| | cPGD | **92.9** | **88.4** | **96.7** | 78.8 | **53.8** | **84.4** | **92.0** | **90.3** | 93.9 | 83.6 | **50.4** | **81.3** | **82.2** |

Table 1: Win rates of preference optimization methods against the SDXL model on the Pick-a-Pic v2 test set and the Parti-Prompts benchmark. Model checkpoints for other methods are provided by their respective authors. The 1st-best results are **bolded** and the 2nd-best results are underlined.

**General results.** As shown in Fig. 5, Table 1 and Table 2, we find that our proposed method PGD and cPGD generally outperform the baselines in achieving higher absolute reward values and win rates for different test prompt sets and different base models. While our methods generally achieve lower Aes scores, the behavior is less indicative because our training objective is to align with the human preference implied by text-image paired datasets, while Aes is an unconditional reward model that does not take text-image alignment into consideration.

**Diversity and prior preservation.** We further demonstrate the tradeoffs between reward, FID (measuring prior preservation) and diversity scores in Fig. 9 and Fig. 8. The blue regions are the combinations that are strictly dominated by the performance of our methods, the boundary of which is formed by the performance resulted from different choices of guidance weights.

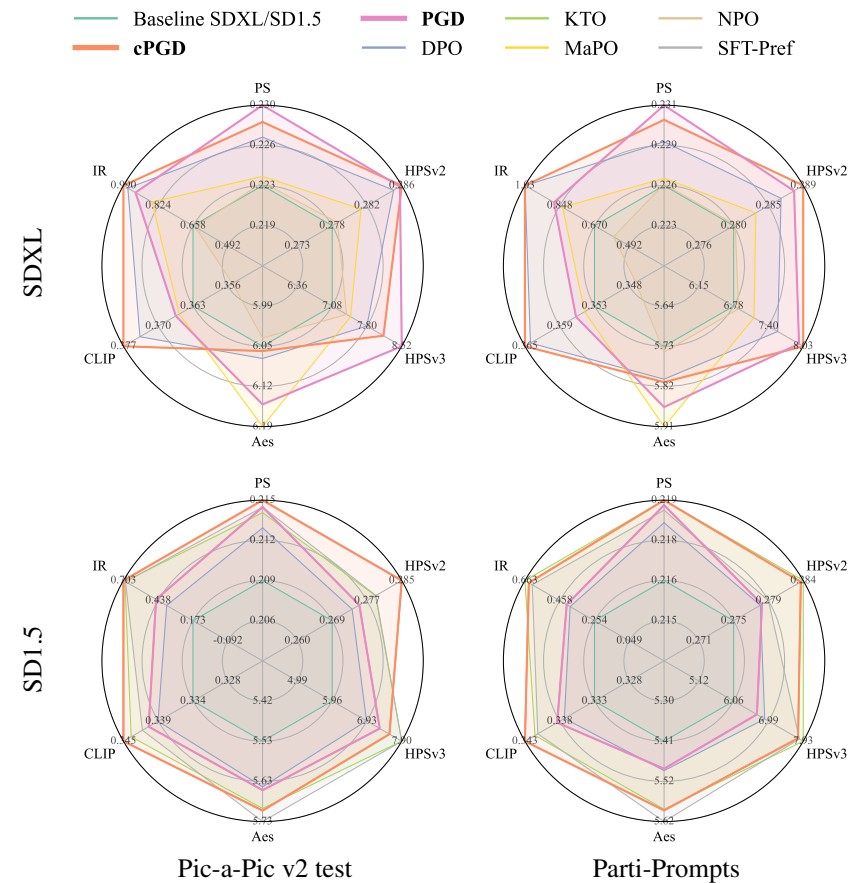

Figure 5: Overall comparison on SDXL (top) and SD1.5 (bottom). Radar axes report mean scores (higher is better): PickScore (PS), HPSv2, HPSv3, Aesthetics (Aes), CLIP, and ImageReward (IR). Polygons closer to the outer rim indicate better aggregate performance across metrics.

| Base Model | Inference Strategy | Pic-a-Pic v2 test (424 prompts) | | | | | | Parti-Prompts (1632 prompts) | | | | | | Avg. ↑ |
|---|---|---|---|---|---|---|---|---|---|---|---|---|---|---|
| | | PS ↑ | HPSv2 ↑ | HPSv3 ↑ | Aes ↑ | CLIP ↑ | IR ↑ | PS ↑ | HPSv2 ↑ | HPSv3 ↑ | Aes ↑ | CLIP ↑ | IR ↑ | |
| SD1.5 | – | 50.0 | 50.0 | 50.0 | 50.0 | 50.0 | 50.0 | 50.0 | 50.0 | 50.0 | 50.0 | 50.0 | 50.0 | 50.0 |
| | PGD | **78.3** | 71.2 | 67.9 | 62.3 | 58.5 | 63.7 | **68.0** | 65.0 | 59.6 | 58.1 | 55.0 | 56.9 | 63.7 |
| | cPGD | 76.9 | **71.7** | **71.9** | **63.2** | **59.9** | **72.2** | 66.4 | **76.9** | **68.9** | **68.1** | **58.4** | **71.0** | **68.8** |
| DPO-SD1.5 | – | 76.4 | 67.7 | 66.3 | 65.1 | 55.9 | 60.6 | 67.3 | 64.8 | 64.5 | 53.6 | 61.0 | 63.8 |
| | PGD | **79.2** | 70.3 | 66.3 | 66.3 | 59.4 | 63.2 | **74.2** | 67.4 | 65.0 | 63.1 | 55.5 | 62.9 | 66.1 |
| | cPGD | 79.0 | **81.8** | **75.5** | **71.7** | **62.3** | **76.2** | 73.8 | **74.1** | **69.6** | **69.0** | **59.0** | **67.3** | **71.6** |
| KTO-SD1.5 | – | 72.6 | 78.1 | 76.2 | 68.6 | 58.5 | **75.0** | 66.6 | 78.3 | 71.9 | 68.8 | 53.3 | 71.3 | 69.9 |
| | PGD | **81.6** | **83.3** | **80.2** | 70.3 | **61.1** | 77.4 | **72.1** | **80.3** | **72.8** | 72.4 | 55.1 | **73.8** | **73.4** |
| | cPGD | 76.7 | 80.2 | 75.9 | **70.5** | 60.4 | 74.3 | 66.2 | 77.1 | 69.5 | 68.4 | **55.9** | 72.2 | 70.6 |
| SPO-SD1.5 | – | 71.2 | 63.0 | 64.9 | 68.2 | 38.7 | 61.1 | 68.6 | 61.2 | 64.2 | 71.9 | 37.7 | 61.6 | 61.0 |
| | PGD | 79.7 | 70.3 | 69.6 | 69.8 | 44.3 | 67.9 | 74.2 | 66.5 | 66.2 | 72.2 | 46.7 | 67.0 | 66.2 |
| | cPGD | **82.3** | **81.8** | **75.5** | **71.2** | **60.6** | **76.2** | **74.8** | **71.9** | **71.3** | **73.9** | **47.5** | **72.9** | **71.7** |

Table 2: Win rates of preference optimization methods against the SD1.5 model on the Pic-a-Pic v2 test set and the Parti-Prompts benchmark. Model checkpoints for other methods are provided by their respective authors. The 1st-best results are **bolded** and the 2nd-best results are underlined.

**PGD vs. cPGD.** We find that cPGD is generally better on SD1.5 but comparable on SDXL. We hypothesize that such behavior is due to the distribution shift between the image distributions of the preference datasets and that of the base model. As the images in the preference datasets we used are generally better than those from SD1.5, the dynamic reweighting mechanism used in cPGD helps generalization in this case.

**Transfer to other base models.** Inspired by the plug-and-play nature of our approach, we experiment with aligning base models that are finetuned with alternative DPO variants on the same preference datasets using our PGD/cPGD-finetuned modules (*e.g.*, the second mega-row "DPO-SDXL" in Table 1 demonstrate the performance when using DPO-tuned SDXL as the inference-time base

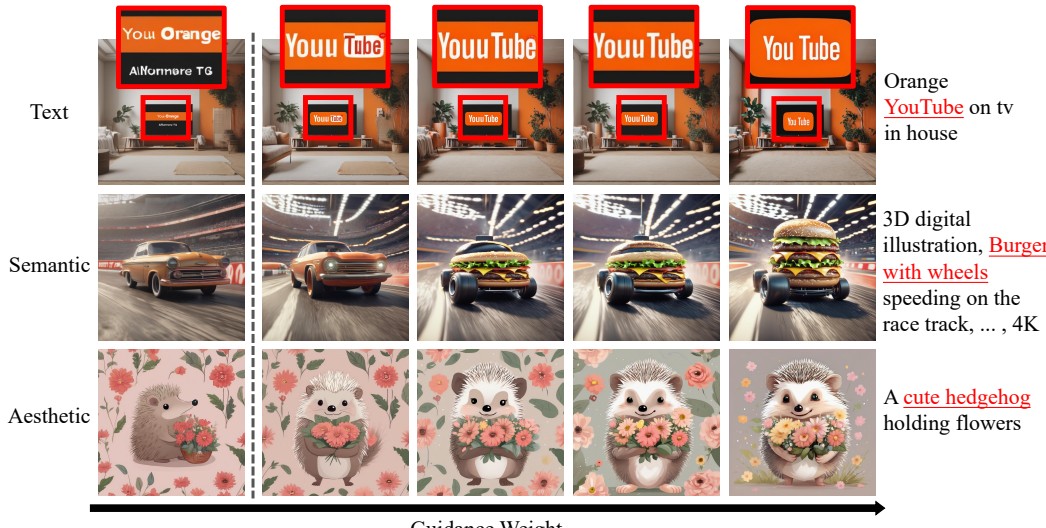

Figure 6: Qualitative effect of increasing guidance weight $w$ (left → right). Rows show text fidelity, semantic binding, and aesthetic style. Stronger $w$ improves alignment and legibility up to a mid range, after which overshooting/rigidity appears.

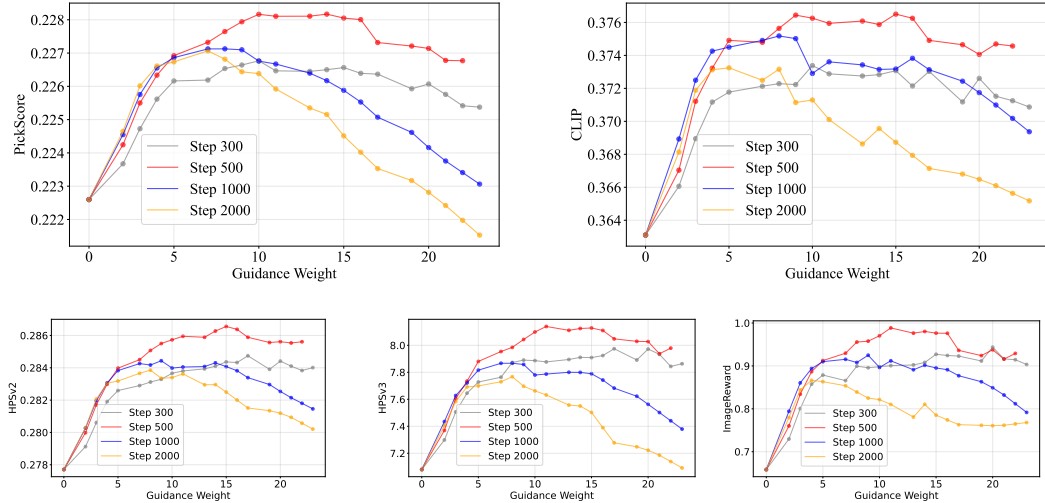

Figure 7: Effect of guidance weight $w$ on automatic metrics (SDXL). Left: PickScore; Right: CLIP score. "Step" denotes the training steps of the guidance module. Curves rise quickly for small $w$,

model). We find that there is nearly consistent improvement compared to any original base model, which is made easy with the CFG-style inference rule in our PGD method.

**Ablation on guidance weights.** Qualitatively, increasing guidance weights generally yields better reward following. as shown in Fig. 6. To comprehensively quantify the effect of guidance weights in different hyperparameter settings, we measuring the performance metrics by varying the guidance weights for models finetuned on Pick-a-Pic v2 from SDXL with 300, 500, 1000 and 2000 steps. As shown in Fig. 7, we observe that increasing guidance weights from 0 to some moderate value (around 6) generally leads to better reward values for all tested models, but beyond that the model performance drops. Models finetuned with less steps exhibit less amount of performance drop, which is likely due to the regularization effect of early stopping. Furthermore, Fig. 9 and 8 shows that 1) the increase of guidance weights leads to "less natural" images and 2) if the guidance weight is beyond certain threshold, increasing the guidance weight leads to more chaotic predictions.

**Dataset quality.** Since the image distribution in the preference datasets can differ from the image distribution of the base models, here we investigate how methods are robust to different preference

Table 3: Impact of preference data variance on alignment performance. "Subset" refers to training on a high-quality curated subset (low variance), while "Fullset" uses the full HPDv3 dataset (high variance). The 1st-best results are in **bold** and the 2nd-best are underlined. All methods are applied to the base SDXL model.

| Set | Method | PS ↑ | HPSv2 ↑ | HPSv3 ↑ | Aes ↑ | CLIP ↑ | IR ↑ |
|-----|--------|------|---------|---------|-------|--------|------|
| Subset | SDXL | 0.2226 | 0.2777 | 7.0795 | 6.0521 | 0.3631 | 0.6583 |
| | DPO | 0.2253 | 0.2828 | 8.0327 | 6.1124 | 0.3619 | 0.8231 |
| | PGD | 0.2257 | 0.2854 | 9.4588 | 6.2029 | 0.3637 | 0.8741 |
| | cPGD | **0.2276** | **0.2889** | **10.0454** | **6.2670** | **0.3646** | **1.0312** |
| Fullset | SDXL | 0.2226 | 0.2777 | 7.0795 | 6.0521 | 0.3631 | 0.6583 |
| | DPO | 0.2266 | 0.2847 | 8.2610 | 6.1658 | 0.3659 | 0.9179 |
| | PGD | **0.2285** | 0.2871 | **10.0649** | **6.5050** | 0.3644 | 1.0625 |
| | cPGD | 0.2273 | **0.2902** | 9.2426 | 6.1791 | **0.3734** | **1.1433** |

datasets, especially when the image quality differs a lot. In Table 3, we show the results of finetuning on the full HPDv3 dataset, in which the variance of image quality is great, and on a high-quality subset. We find that our methods generally performs better in both cases, but on the high-quality subset our methods unanimously outperform the baselines. In addition, in the high-quality subset case, cPGD is unanimously better than PGD. We hypothesize that this is likely due to that cPGD imposes weaker assumptions on preference pairs as $\theta^+$ and $\theta^-$ are trained in an independent way. Despite that the high-quality subset yields more consistent observations, using the full dataset generally leads to better reward values, in part simply due to the increased number of data points.

## 6 DISCUSSIONS

**PGD as kernel method.** As shown in our experiments, PGD inference with slightly finetuned models consistently outperforms DPO methods. This behavior can be understood through the theory of neural tangent kernels (NTK) (Jacot et al., 2018). In the *lazy training regime*, i.e. when the finetuned model remains close to the reference model, we can write $\epsilon_{\text{ref}} + w(\epsilon_{\text{finetuned}} - \epsilon_{\text{ref}}) \approx \epsilon_{\text{ref}} + wK_{\text{ref}}\alpha$ where $K_{\text{ref}}$ is known as the NTK matrix of $\epsilon_{\text{ref}}$ and $\alpha$ is the vector of regression coefficients, which shows that PGD is essentially kernel regression in the NTK feature space of the reference model. Because this feature space is an intrinsic and stable property of $\epsilon_{\text{ref}}$, PGD inference leverages reliable features. In contrast, extended finetuning with large learning rates can push the model out of the lazy regime, causing the NTK approximation to break down and increasing the risk of overfitting on small datasets.

**Inference cost.** While the inference time is doubled with PGD due to the need to compute outputs with the reference model, we note that it is possible to perform distillation so that one single model learns to predict the PGD outputs, as demonstrated by many other works on diffusion model distillation (Salimans & Ho, 2022; Song et al., 2023; Meng et al., 2023). To verify this, we present our attempts in distilling a single model out of cPGD in Appendix A.4.

## 7 CONCLUSION

We introduced preference-guided diffusion (PGD), a simple yet effective method that better aligns diffusion models with human preference through the lens of classifier-free guidance: the finetuned model is the guidance signal of the dataset. By further take inspiration from the training of conditional diffusion models, we propose a variant called contrastive PGD (cPGD) which parameterize the finetuned module with two models independently trained on positive and negative samples, respectively. We empirically verify the effectiveness of the proposed methods on different datasets and base models.

ETHICS STATEMENT

This work investigates preference-guided generation for text-to-image diffusion models. We did not collect new human subjects data; all experiments use publicly available datasets and prompt suites (e.g., Pick-a-Pic v2, HPDv2/3, Parti-Prompts) under their respective licenses and intended-use policies. No personally identifiable information (PII) was processed to our knowledge. Where dataset curators provide safety filters or content flags, we follow them and do not intentionally prompt for unsafe content.

Preference signals and reward models may reflect societal biases (e.g., aesthetics tied to culture, gender, or geography). Such biases can be amplified by guidance at inference time. We therefore (i) report multiple metrics, including diversity and base-model fidelity, (ii) encourage down-stream deployers to pair our method with content filters, auditing on representative user groups, and opt-out mechanisms, and (iii) commit to releasing prompts, seeds, and code sufficient for re-producibility while avoiding lists or examples that facilitate misuse (e.g., targeted impersonation or non-consensual content).

Finally, we oppose harmful uses of generative models (e.g., harassment, disinformation, infringement) and will abide by takedown requests from dataset owners within their policies.

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

# A APPENDIX

## A.1 FURTHER IMPLEMENTATION DETAILS

**Software and precision.** All models are implemented in `PyTorch` with `diffusers` and memory–efficient attention (`xFormers`). Mixed precision (fp16/bf16) is enabled for both training and inference.

**Sampling and guidance.** Unless otherwise noted, SDXL uses the default 50-step DDIM (Song et al., 2020) sampler; for fairness, SD1.5 is also run with 50 steps. Text-to-image generation employs classifier-free guidance (CFG), with the CFG weight fixed to 5.0 for SDXL and 7.5 for SD1.5. All other sampling hyperparameters are kept identical across methods.

**Training protocol.** Unless noted, we follow the optimizer choices and step counts in Sec. 5.1 (DPO/PGD main experiments: 2k training steps; distilled/cPGD ablations: 500 training steps). VAE weights are kept frozen; all methods share identical image resolutions and augmentations (random resize–crop and horizontal flip with probability 0.5).

**HPDv3 subset construction.** To obtain a *curated* low-variance (in image quality) subset, we restrict HPDv3 to pairs whose two candidate images both have recorded provenance in {`flux`, `kolors`, `sd3`, `sdxl`, `hunyuan`, `real_images`, `infinity`, `midjourney`}. Pairs are discarded if either side falls outside this list or has unknown origin.

**Evaluation protocol.** We compute PickScore, HPSv2, HPSv3, ImageReward, CLIP Score, and Aesthetics (Aes) using their public checkpoints and each model's default preprocessing. Win rate is the percentage of prompts for which a method's image scores higher than the base model under identical prompts, sampler settings, and a fixed random seed; for win rate, we generate one image per prompt per method. FID and CLIP-Diversity follow Appendix *Computation of Evaluation Metrics*: we sample $K{=}25$ prompts and, for each prompt, generate $n{=}40$ images using 40 distinct seeds, then aggregate across the $K$ prompts.

## A.2 COMPUTATION OF EVALUATION METRICS

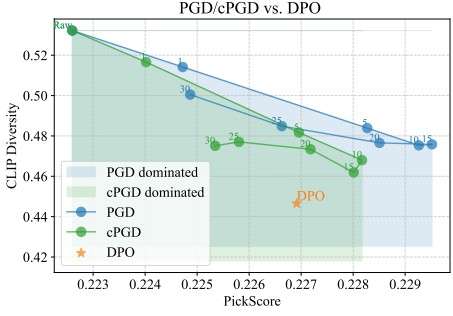

Figure 8: Reward–diversity Pareto.

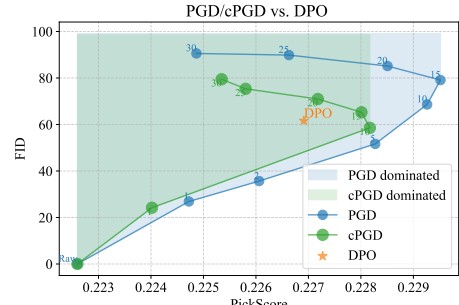

Figure 9: Reward–fidelity Pareto.

**Diversity.** To quantify the diversity of generated images, we adopt the CLIP-based diversity score (Domingo-Enrich et al., 2025; Liu et al., 2025). This metric measures the variance within a set of images generated from a single prompt. Formally, for the $k$-th prompt, we generate $n{=}40$ images $\{g_i^k\}_{i=1}^{40}$ using the `open_clip` encoder $\phi(\cdot)$ (Cherti et al., 2023), and average the pairwise squared $\ell_2$ distances of their embeddings. The final diversity score is then averaged over $K{=}25$ distinct prompts:

$$\text{CLIP-Diversity} = \frac{1}{K}\sum_{k=1}^{K}\frac{2}{n(n-1)}\sum_{1\leq i<j\leq n}\left\|\phi(g_i^k)-\phi(g_j^k)\right\|_2^2$$

**FID.** We also report Fréchet Inception Distance (FID; Heusel et al. (2017)) as a measure of distributional shift to the SDXL. The two feature distributions are: (i) embeddings of images generated by the SDXL reference using the same prompt set and default sampling settings; and (ii) embeddings of

images generated by the method under evaluation (e.g., DPO, PGD variants) with identical prompts and seeds. We extract 2048-d pool3 activations from a pre-trained Inception-V3 after standard pre-processing (resize/crop to $299\times299$ and Inception normalization), and compute FID between the two empirical distributions.

### A.3 HUMAN PREFERENCE STUDY

**Setup and protocol.** We conduct a human preference evaluation using a bilingual (English/Chinese) web interface; instructions are shown in both languages with English taking precedence. For each question, we display six images generated from the *same* text prompt by six systems (Raw, DPO, MaPO, NPO, PGD, cPGD). Images are *anonymized* and labeled only by random letters (e.g., P/M/C/R/D/N). To mitigate order and identity biases, (i) the six thumbnails are randomly permuted per question, and (ii) the letter–method mapping is re-sampled for every question (hence letters are not consistent across questions). Participants are instructed to **select 1–3 images** they prefer according to the stated criteria: *Text Alignment* (faithfulness to the prompt text), *Semantic Alignment* (scene/logic consistency), and *Aesthetic Preference* (composition, lighting, style).

**Subjects and materials.** We evaluate 55 prompts with 20 participants, yielding $55\times20=1100$ prompt–participant decision units. Because multiple selections are allowed, the total number of recorded choices ("votes") exceeds the number of questions: we collect 1848 valid votes in total ($\approx1.68$ selections per decision unit; $\approx33.6$ selections per prompt).

**Aggregation metric.** Each checked option contributes one vote to the corresponding method. We report the *vote share* of each method, defined as its total votes divided by the grand total of votes across all methods.

| Method | Votes | Vote share (%) |
|--------|-------|----------------|
| Raw    | 208   | 18.9 |
| DPO    | 334   | 30.4 |
| MaPO   | 197   | 17.9 |
| NPO    | 256   | 23.3 |
| PGD    | **500** | **45.5** |
| cPGD   | 353 | 32.1 |
| Total  | 1848  | 168.0 |

Table 4: **Summary of human preference votes.** Participants could select 1–3 images per question, hence the total number of votes exceeds the number of questions.

**Results.** As shown in Table 4, PGD achieves the highest vote share (27.1%), a relative improvement of +49.7% over DPO (18.1%). cPGD attains 19.1%, a +5.7% relative gain over DPO. Raw and MaPO obtain 11.3% and 10.7%, respectively. These aggregates are reported at the vote level; per-prompt paired analyses and uncertainty estimates (e.g., bootstrap confidence intervals and sign tests) are provided in the appendix.

### A.4 ADDITION ABLATION ATUDIES

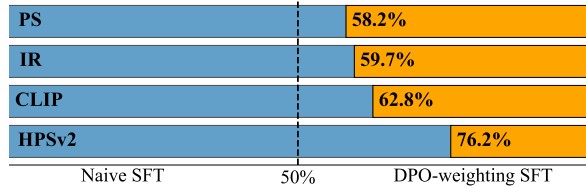

Figure 10: Comparison of SFT training with and without $w_{\mathrm{DPO}}$ weighting. Blue bars indicate naive SFT, and orange bars indicate weighted SFT. All results are obtained under the cPGD inference strategy with the same guidance weight.

**Effect of $w_{\text{DPO}}$ weighting For cPGD**

As shown in Fig. 10, re-weighting the SFT objective with $w_{\text{DPO}}$ consistently lifts win rates above the 50% chance level under the same cPGD guidance weight—58.2% on PickScore, 59.7% on ImageReward, 62.8% on CLIP, and 76.2% on HPSv2. In contrast, naïve SFT hovers around 50%, indicating that uniform supervision underutilizes the signal in preference data. The DPO-derived weights emphasize examples with larger preference margins, aligning the SFT update with the "preference direction" exploited at inference and yielding uniform gains without changing the sampler, prompts, or seeds.

**Partial-step guidance: compute-quality trade-off**

**Setup.** Under *cPGD*, we keep the default SDXL sampler at 50 diffusion steps and apply guidance only to the first $s$ high-noise steps, falling back to the base update afterwards:

$$w_t = \begin{cases} w, & t \leq s, \\ 0, & t > s, \end{cases} \qquad t = 1, \ldots, 50.$$

We report PickScore win rate (vs. SDXL-base, higher is better) and the *time ratio* measured as wall-clock relative to vanilla SDXL 50-step sampling (i.e., 1.0 equals unguided SDXL).

Table 5: Guiding only the first $s$ of 50 steps on SDXL (cPGD).

| Guided steps $s$ | 10 | 20 | 30 | 40 | 50 |
|---|---|---|---|---|---|
| **PickScore win rate (%)** | 70.2 | 72.4 | 74.8 | 76.9 | 79.1 |
| **Time ratio ($\times$ vs. SDXL)** | 1.4 | 1.8 | 2.2 | 2.6 | 3.0 |

**Findings.** Table 5 exhibits a smooth Pareto frontier: as $s$ increases, reward improves monotonically while compute grows roughly linearly. Notably, *even guiding only the first 10 steps* already delivers a strong gain (70.2% win rate) at just $1.4\times$ the cost of vanilla SDXL. Moreover, $s{=}30$ recovers about $94.6\%$ of the full 50-step cPGD reward (74.8 vs. 79.1) at only $\sim 73\%$ of its compute (2.2 vs. 3.0), and $s{=}40$ reaches $97.2\%$ of full reward at $\sim 87\%$ of the compute. In practice, selecting $s$ around 30–40 provides a favorable balance between preference adherence and efficiency, while $s{=}10$ is a compelling low-cost setting when latency is critical.

**Distillation.** To improve efficiency, we compress our guidance into a single checkpoint via a simple distillation procedure. Concretely, on Pick-a-Pic v2 we train for 500 steps using the standard $\epsilon$-prediction loss

$$\mathcal{L}_{\text{distill}} = \mathbb{E}_{(x_0, c), t, \epsilon} \left[ \, \| \, \hat{\epsilon} - \epsilon_\phi(x_0, \epsilon, t, c) \, \|_2^2 \right],$$

where $\hat{\epsilon}$ is obtained from the cPGD guidance formula. We then evaluate the distilled cPGD checkpoint against the raw SDXL base model and a DPO-tuned SDXL on three preference proxies (PickScore, HPSv3, and ImageReward). Figure 11 reports *win rates* (%), measured as the fraction of prompts for which a method outperforms the SDXL base under identical seeds. The distilled model consistently surpasses DPO on PickScore and HPSv3 while remaining competitive on ImageReward, demonstrating that the benefits of preference guidance can be preserved even when compressed into a single offline checkpoint.

## A.5 USE OF LARGE LANGUAGE MODELS (LLMS)

We used a commercial LLM (ChatGPT) as a general-purpose assistant for copy-editing, wording refinement, and LaTeX formatting (e.g., table/figure placement, caption polishing), and for generating minor boilerplate code (argument parsing and plotting stubs) that we subsequently verified and edited. The LLM was *not* used for research ideation, core algorithm design or implementation, dataset construction/labeling, experiment selection, quantitative analysis, or to produce any reported results or images. All scientific claims and conclusions were written and validated by the authors, who take full responsibility for the content—including any LLM-assisted passages—and for proper citation to primary sources. The LLM is not an author.

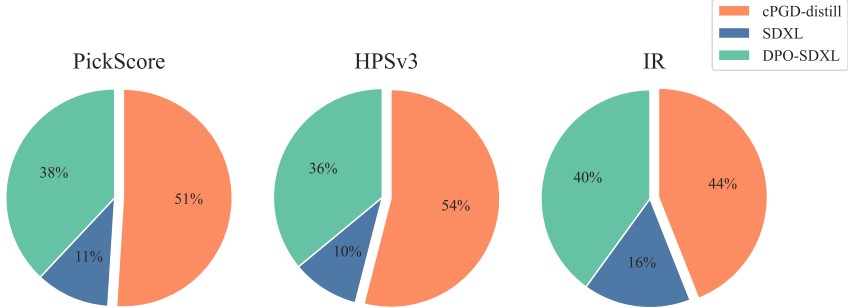

Figure 11: Win rate (%) on PickScore (PS), HPSv3, and ImageReward when the three models—Raw (the original SDXL base), DPO, and cPGD-distill—are compared jointly. For each prompt (same random seeds), we score images from all three models and report the proportion on which each model achieves the highest score (higher is better).The distilled cPGD (500 steps) surpasses DPO on PickScore and HPSv3 and remains competitive on ImageReward.

Table 6: Win rates of preference optimization methods against the SDXL model on the Pick-a-Pic v2 test set benchmark. Model checkpoints for other methods are provided by their respective authors.

| Base Model | Inference Strategy | PS ↑ | HPSv2 ↑ | HPSv3 ↑ | Aes ↑ | CLIP ↑ | IR ↑ | Average ↑ |
|---|---|---|---|---|---|---|---|---|
| SDXL | – | 50.00% | 50.00% | 50.00% | 50.00% | 50.00% | 50.00% | 50.00% |
| | cPGD | 79.95% | 80.19% | 77.12% | 50.94% | 64.86% | 69.81% | 70.48% |
| | PGD | 78.77% | 79.01% | 73.58% | 51.89% | 63.68% | 69.34% | 69.38% |
| | NPO | 58.73% | 59.20% | 69.10% | 52.12% | 37.50% | 53.54% | 55.03% |
| DPO-SDXL | – | 71.70% | 77.59% | 67.92% | 53.30% | 61.56% | 65.80% | 66.31% |
| | cPGD | 80.90% | 77.59% | 84.67% | 63.92% | 58.73% | 64.86% | 71.78% |
| | PGD | 83.25% | 85.38% | 85.61% | 59.67% | 62.26% | 73.58% | 74.96% |
| | NPO | 76.89% | 81.84% | 81.37% | 53.77% | 57.78% | 70.75% | 70.40% |
| MaPO-SDXL | – | 55.90% | 65.33% | 61.79% | 68.16% | 50.24% | 68.16% | 61.60% |
| | cPGD | 77.36% | 78.77% | 72.88% | 69.10% | 59.43% | 72.41% | 71.66% |
| | PGD | 80.42% | 81.60% | 81.60% | 75.71% | 51.42% | 72.17% | 73.82% |
| SPO-SDXL | – | 89.39% | 83.02% | 95.99% | 81.84% | 33.25% | 78.77% | 77.04% |
| | cPGD | 92.92% | 88.44% | 96.70% | 78.77% | 53.77% | 84.43% | 82.51% |
| | PGD | 92.22% | 86.08% | 96.46% | 82.08% | 42.45% | 81.37% | 80.11% |

## A.6 FULL QUANTITATIVE RESULTS

We report complete per-metric win rates (WR, %) across three prompt suites for both SDXL and SD 1.5. Results on SDXL with Pick-a-Pic v2, HPDv2, and Parti-Prompts appear in Tables 6–8; SD 1.5 results are given in Tables 9–11. Across all benchmarks, our guided variants outperform baselines on preference-aligned proxies (PickScore, HPSv2/v3, ImageReward), yielding higher unweighted *average* win rate.

Within each block, the row marked "–" is the reference policy evaluated against itself and therefore equals $50\%$ by construction. Numbers above $50\%$ indicate systematic wins over the base under identical prompts, sampler settings, and fixed seed. The *Average* column is the unweighted mean over PS, HPSv2, HPSv3, Aes, CLIP, and IR.

**High-level trends.** (1) On SDXL, both PGD and cPGD consistently lift the underlying models across all three suites, with larger gains when guiding stronger preference-tuned bases (e.g., DPO/MaPO/SPO). (2) On SD 1.5, the same pattern holds: PGD brings uniform improvements across metrics, while cPGD recovers most of the PGD gains with lower compute. (3) Improvements concentrate on PS/HPS/IR, while Aes/CLIP remain competitive—consistent with guidance emphasizing the learned preference direction rather than style drift.

Table 7: Win rates of preference optimization methods against the SDXL model on the HPD v2 test set benchmark. Model checkpoints for other methods are provided by their respective authors.

| Base Model | Inference Strategy | PS ↑ | HPSv2 ↑ | HPSv3 ↑ | Aes ↑ | CLIP ↑ | IR ↑ | Average ↑ |
|---|---|---|---|---|---|---|---|---|
| SDXL | – | 50.00% | 50.00% | 50.00% | 50.00% | 50.00% | 50.00% | 50.00% |
| | cPGD | 76.50% | 75.50% | 71.25% | 52.25% | 59.00% | 69.00% | 67.25% |
| | PGD | 85.50% | 73.50% | 78.75% | 69.25% | 46.75% | 73.00% | 71.13% |
| | NPO | 63.75% | 65.00% | 78.75% | 60.50% | 42.75% | 55.50% | 61.04% |
| DPO-SDXL | – | 66.75% | 73.75% | 66.00% | 59.75% | 55.25% | 73.00% | 65.75% |
| | cPGD | 75.25% | 79.50% | 71.25% | 57.75% | 61.00% | 73.00% | 69.63% |
| | PGD | 84.25% | 85.50% | 82.25% | 66.75% | 55.00% | 75.25% | 74.83% |
| | NPO | 78.00% | 86.00% | 78.00% | 60.75% | 53.25% | 76.25% | 72.04% |
| MaPO-SDXL | – | 55.75% | 72.50% | 68.00% | 69.25% | 50.75% | 65.50% | 63.63% |
| | cPGD | 73.75% | 81.25% | 77.00% | 67.75% | 57.00% | 70.75% | 71.25% |
| | PGD | 86.50% | 79.25% | 83.75% | 74.25% | 50.25% | 73.00% | 74.50% |
| SPO-SDXL | – | 93.00% | 90.00% | 97.25% | 86.00% | 32.75% | 77.25% | 79.38% |
| | cPGD | 93.25% | 91.75% | 97.00% | 81.25% | 42.75% | 81.25% | 81.21% |
| | PGD | 95.00% | 90.75% | 96.00% | 84.50% | 34.50% | 77.25% | 79.67% |

Table 8: Win rates of preference optimization methods against the SDXL model on the Parti-Prompts set benchmark. Model checkpoints for other methods are provided by their respective authors.

| Base Model | Inference Strategy | PS ↑ | HPSv2 ↑ | HPSv3 ↑ | Aes ↑ | CLIP ↑ | IR ↑ | Average ↑ |
|---|---|---|---|---|---|---|---|---|
| SDXL | – | 50.00% | 50.00% | 50.00% | 50.00% | 50.00% | 50.00% | 50.00% |
| | cPGD | 75.80% | 80.27% | 77.82% | 57.23% | 62.01% | 74.02% | 71.19% |
| | PGD | 78.74% | 78.37% | 78.19% | 68.75% | 53.98% | 69.06% | 71.18% |
| | NPO | 55.02% | 56.37% | 62.07% | 55.09% | 38.97% | 51.53% | 53.18% |
| DPO-SDXL | – | 63.97% | 70.28% | 64.03% | 57.54% | 58.46% | 69.73% | 64.00% |
| | cPGD | 73.90% | 79.23% | 72.79% | 59.38% | 64.09% | 74.45% | 70.64% |
| | PGD | 80.76% | 83.88% | 81.74% | 67.65% | 57.48% | 76.10% | 74.60% |
| | NPO | 70.59% | 78.37% | 77.51% | 63.36% | 56.25% | 70.83% | 69.49% |
| MaPO-SDXL | – | 52.02% | 64.40% | 58.52% | 72.37% | 48.22% | 65.01% | 60.09% |
| | cPGD | 72.49% | 81.13% | 76.90% | 70.10% | 58.21% | 74.39% | 72.20% |
| | PGD | 78.86% | 77.82% | 79.60% | 77.76% | 53.55% | 72.67% | 73.38% |
| SPO-SDXL | – | 87.81% | 85.48% | 92.16% | 88.11% | 31.74% | 74.94% | 76.71% |
| | cPGD | 91.97% | 90.32% | 93.87% | 83.64% | 50.37% | 81.31% | 81.91% |
| | PGD | 91.36% | 87.75% | 93.87% | 88.36% | 47.98% | 77.33% | 81.11% |

Table 9: Win rates of preference optimization methods against the SD1.5 model on the Pick-a-Pic v2 test set benchmark. Model checkpoints for other methods are provided by their respective authors.

| Base Model | Inference Strategy | PS ↑ | HPSv2 ↑ | HPSv3 ↑ | Aes ↑ | CLIP ↑ | IR ↑ | Average ↑ |
|---|---|---|---|---|---|---|---|---|
| SD1.5 | – | 50.00% | 50.00% | 50.00% | 50.00% | 50.00% | 50.00% | 50.00% |
| | cPGD | 76.89% | 71.70% | 71.93% | 63.21% | 59.91% | 72.17% | 69.30% |
| | PGD | 78.30% | 71.23% | 67.92% | 62.26% | 58.49% | 63.68% | 66.98% |
| DPO-SD1.5 | – | 76.42% | 67.69% | 66.27% | 65.09% | 55.90% | 60.61% | 65.33% |
| | cPGD | 79.01% | 81.84% | 75.47% | 71.70% | 62.26% | 76.18% | 74.41% |
| | PGD | 79.25% | 70.28% | 66.27% | 66.27% | 59.43% | 63.21% | 67.45% |
| KTO-SD1.5 | – | 72.64% | 78.07% | 76.18% | 68.63% | 58.49% | 75.00% | 71.50% |
| | cPGD | 76.65% | 80.19% | 75.94% | 70.51% | 60.38% | 74.29% | 72.99% |
| | PGD | 81.60% | 83.25% | 80.19% | 70.28% | 61.08% | 77.36% | 75.63% |
| SPO-SD1.5 | – | 71.23% | 62.97% | 64.86% | 68.16% | 38.68% | 61.08% | 61.16% |
| | cPGD | 82.31% | 81.84% | 75.47% | 71.23% | 60.61% | 76.18% | 74.61% |
| | PGD | 79.72% | 70.28% | 69.58% | 69.81% | 44.34% | 67.92% | 66.94% |

Table 10: Win rates of preference optimization methods against the SD1.5 model on the HPD v2 test set benchmark. Model checkpoints for other methods are provided by their respective authors.

| Base Model | Inference Strategy | PS ↑ | HPSv2 ↑ | HPSv3 ↑ | Aes ↑ | CLIP ↑ | IR ↑ | Average ↑ |
|---|---|---|---|---|---|---|---|---|
| SD1.5 | – | 50.00% | 50.00% | 50.00% | 50.00% | 50.00% | 50.00% | 50.00% |
|  | cPGD | 80.50% | 88.00% | 84.00% | 73.00% | 61.25% | 79.50% | 77.71% |
|  | PGD | 75.25% | 68.00% | 67.00% | 67.25% | 52.25% | 65.50% | 65.88% |
| DPO-SD1.5 | – | 76.25% | 69.75% | 68.00% | 66.50% | 57.50% | 65.25% | 67.21% |
|  | cPGD | 80.25% | 81.00% | 79.00% | 70.25% | 60.75% | 75.75% | 74.50% |
|  | PGD | 79.00% | 73.75% | 71.75% | 66.00% | 54.75% | 68.50% | 68.96% |
| KTO-SD1.5 | – | 73.50% | 85.00% | 85.00% | 69.50% | 58.75% | 77.75% | 74.92% |
|  | cPGD | 77.00% | 86.50% | 82.25% | 68.00% | 58.00% | 79.50% | 75.21% |
|  | PGD | 80.50% | 88.50% | 84.75% | 71.75% | 58.75% | 80.50% | 77.46% |
| SPO-SD1.5 | – | 77.25% | 72.75% | 74.00% | 73.75% | 29.25% | 63.25% | 65.04% |
|  | cPGD | 80.75% | 78.75% | 82.25% | 71.25% | 44.50% | 75.00% | 72.08% |
|  | PGD | 81.75% | 74.25% | 75.00% | 75.25% | 35.00% | 67.00% | 68.04% |

Table 11: Win rates of preference optimization methods against the SD1.5 model on the Parti-Prompts set benchmark. Model checkpoints for other methods are provided by their respective authors.

| Base Model | Inference Strategy | PS ↑ | HPSv2 ↑ | HPSv3 ↑ | Aes ↑ | CLIP ↑ | IR ↑ | Average ↑ |
|---|---|---|---|---|---|---|---|---|
| SD1.5 | – | 50.00% | 50.00% | 50.00% | 50.00% | 50.00% | 50.00% | 50.00% |
|  | cPGD | 66.36% | 76.90% | 68.87% | 68.14% | 58.39% | 71.02% | 68.28% |
|  | PGD | 68.01% | 65.01% | 59.56% | 58.15% | 54.96% | 56.86% | 60.43% |
| DPO-SD1.5 | – | 67.34% | 64.83% | 64.46% | 62.19% | 53.55% | 61.03% | 62.23% |
|  | cPGD | 73.84% | 74.08% | 69.61% | 69.00% | 59.01% | 67.34% | 68.81% |
|  | PGD | 74.20% | 67.40% | 65.01% | 63.05% | 55.51% | 62.87% | 64.68% |
| KTO-SD1.5 | – | 66.61% | 78.31% | 71.94% | 68.75% | 53.31% | 71.26% | 68.36% |
|  | cPGD | 66.24% | 77.14% | 69.49% | 68.44% | 55.88% | 72.18% | 68.23% |
|  | PGD | 72.12% | 80.33% | 72.79% | 72.43% | 55.15% | 73.84% | 71.11% |
| SPO-SD1.5 | – | 68.63% | 61.21% | 64.15% | 71.94% | 37.75% | 61.64% | 60.89% |
|  | cPGD | 74.82% | 71.94% | 71.32% | 73.90% | 47.49% | 72.86% | 68.72% |
|  | PGD | 74.20% | 66.54% | 66.24% | 72.24% | 46.69% | 67.03% | 65.49% |

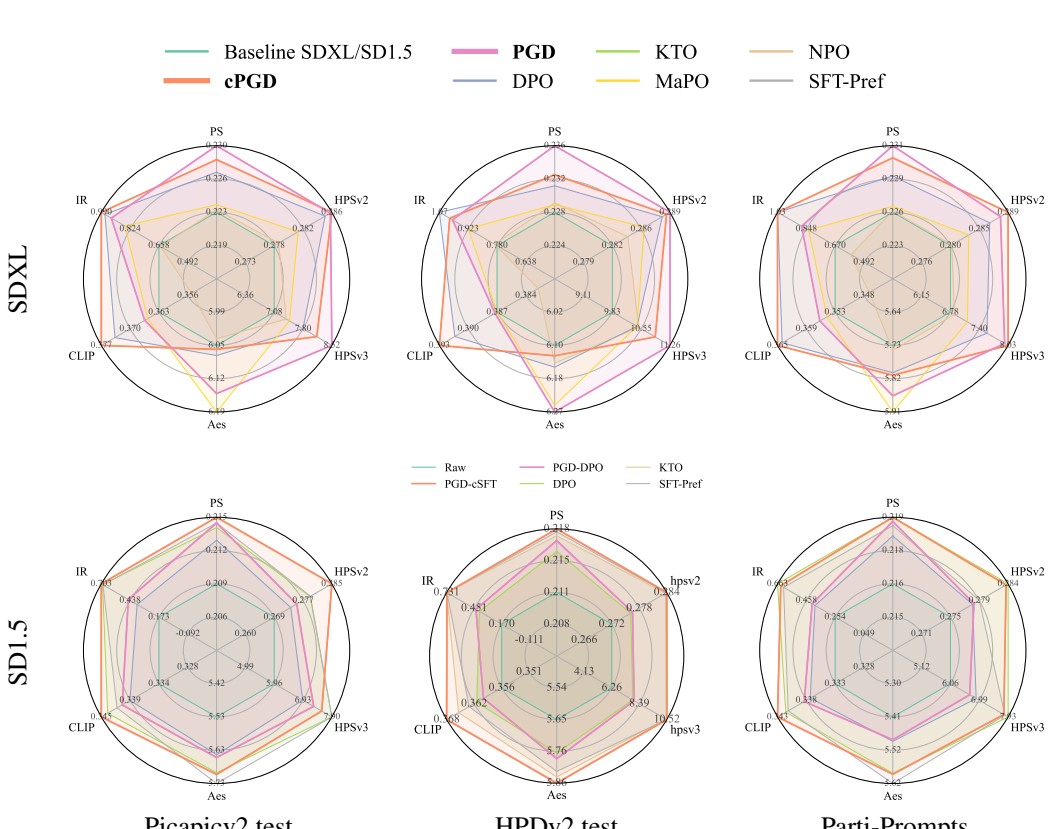

Figure 12: General Results with SDXL, SD1.5.

## A.7 ADDITIONAL QUALITATIVE RESULTS

Figures 13–18 provide additional side-by-side comparisons for PGD and cPGD against representative baselines across varied prompts.

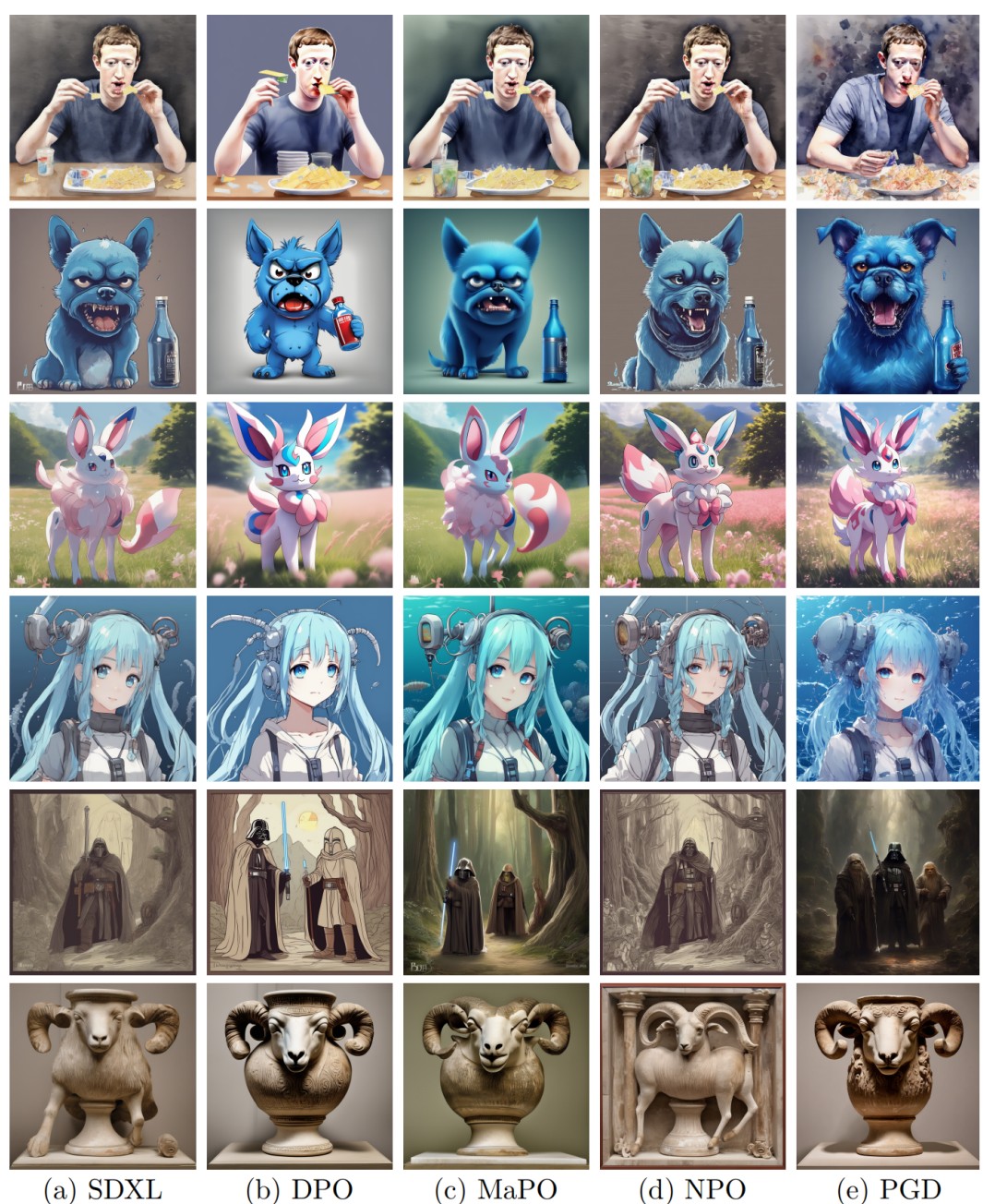

(a) SDXL    (b) DPO    (c) MaPO    (d) NPO    (e) PGD

Figure 13: Comparison of preference-optimization methods on SDXL. Columns show outputs from the base model (SDXL), DPO, MaPO, NPO and PGD.

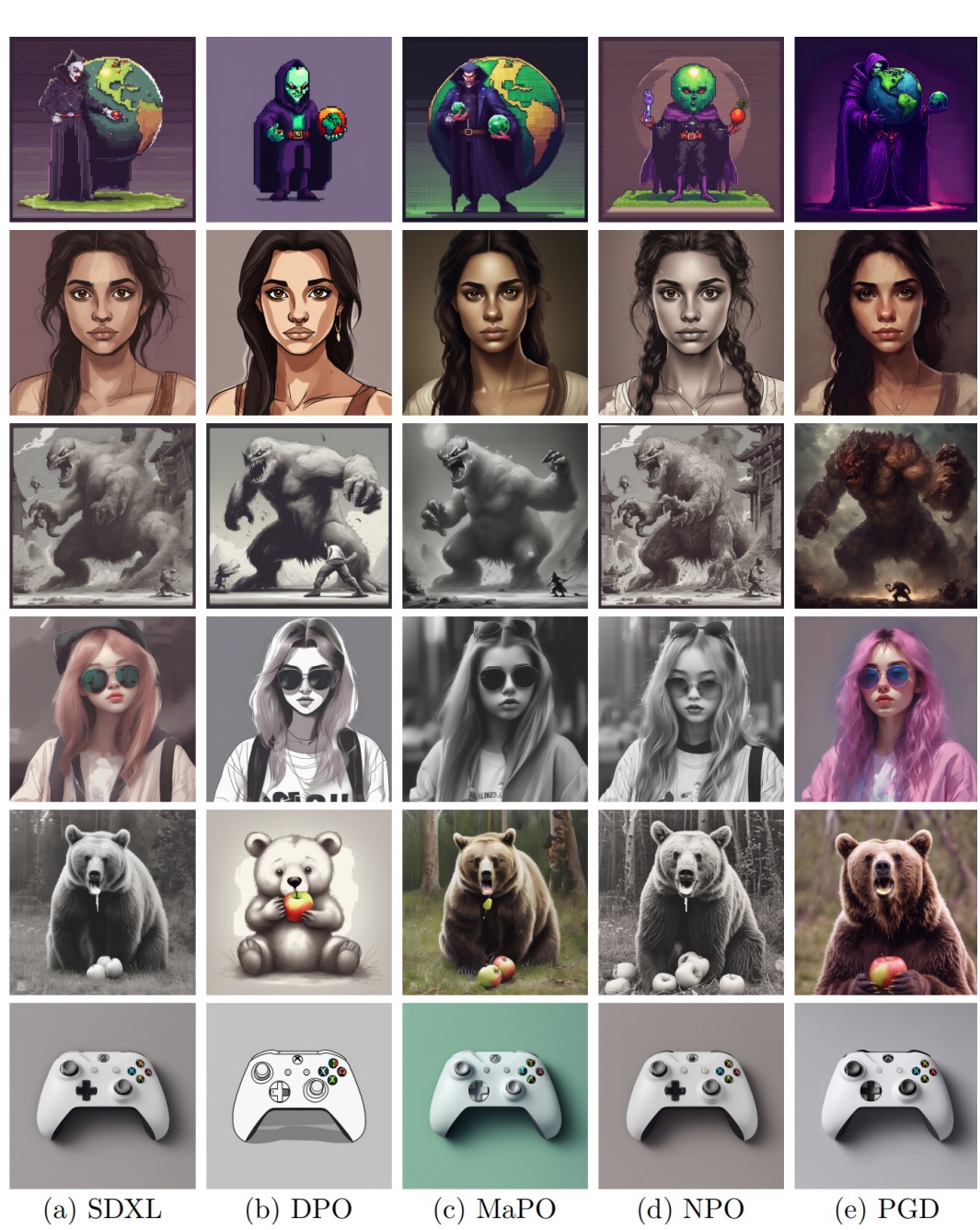

(a) SDXL   (b) DPO   (c) MaPO   (d) NPO   (e) PGD

Figure 14: Comparison of preference-optimization methods on SDXL. Columns show outputs from the base model (SDXL), DPO, MaPO, NPO and PGD.

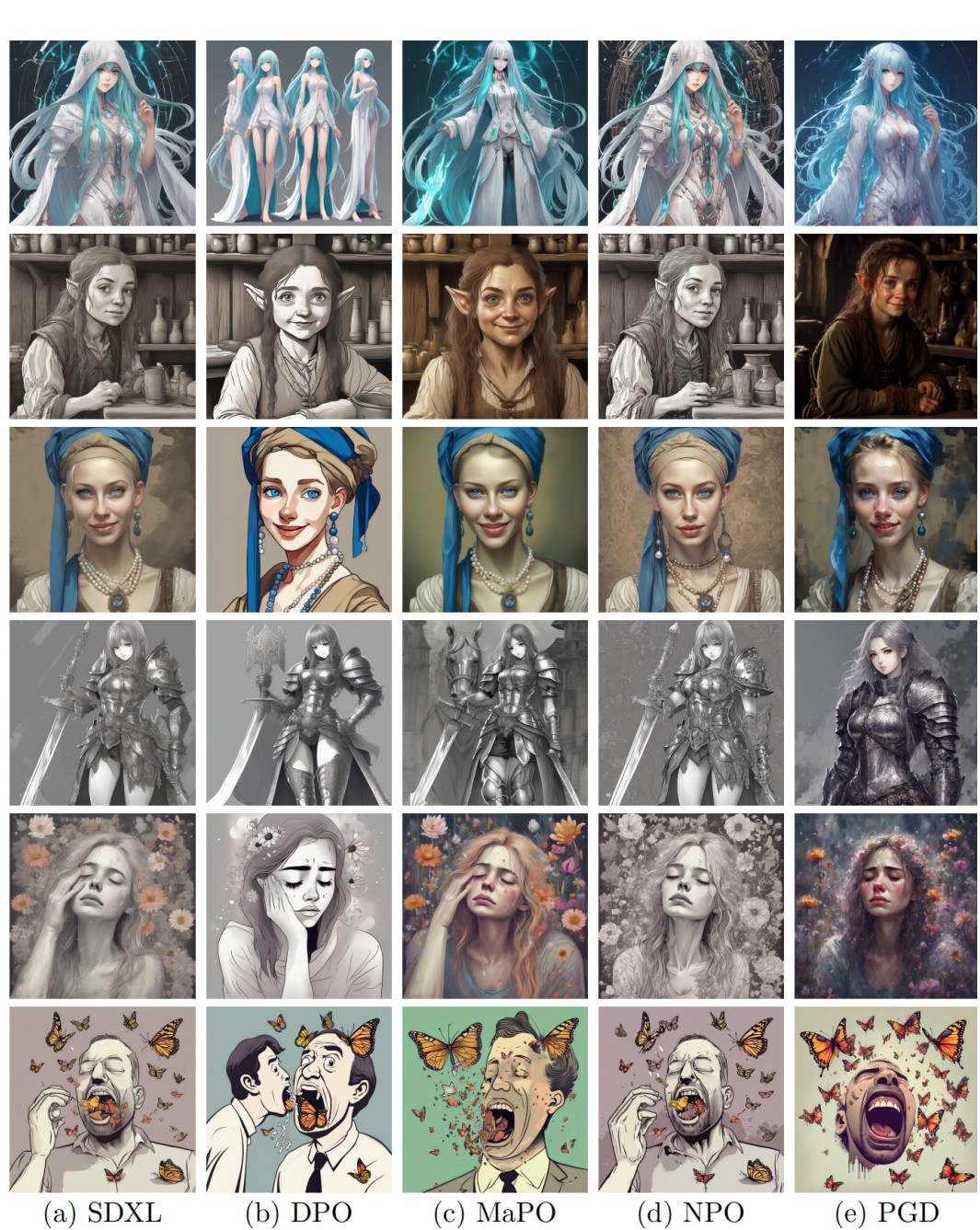

(a) SDXL  (b) DPO  (c) MaPO  (d) NPO  (e) PGD

Figure 15: Comparison of preference-optimization methods on SDXL. Columns show outputs from the base model (SDXL), DPO, MaPO, NPO and PGD.

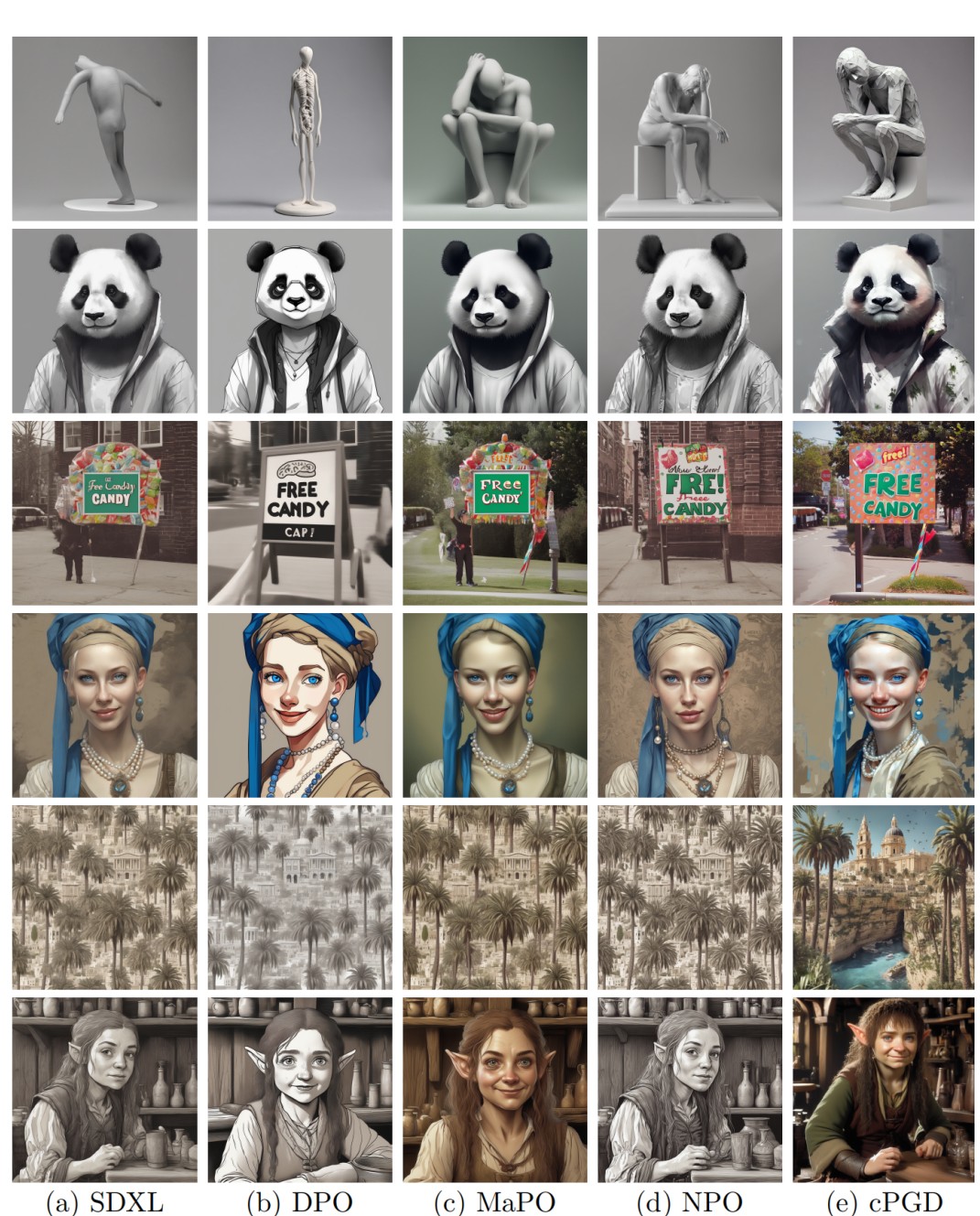

(a) SDXL      (b) DPO      (c) MaPO      (d) NPO      (e) cPGD

Figure 16: Comparison of preference-optimization methods on SDXL. Columns show outputs from the base model (SDXL), DPO, MaPO, NPO and cPGD.

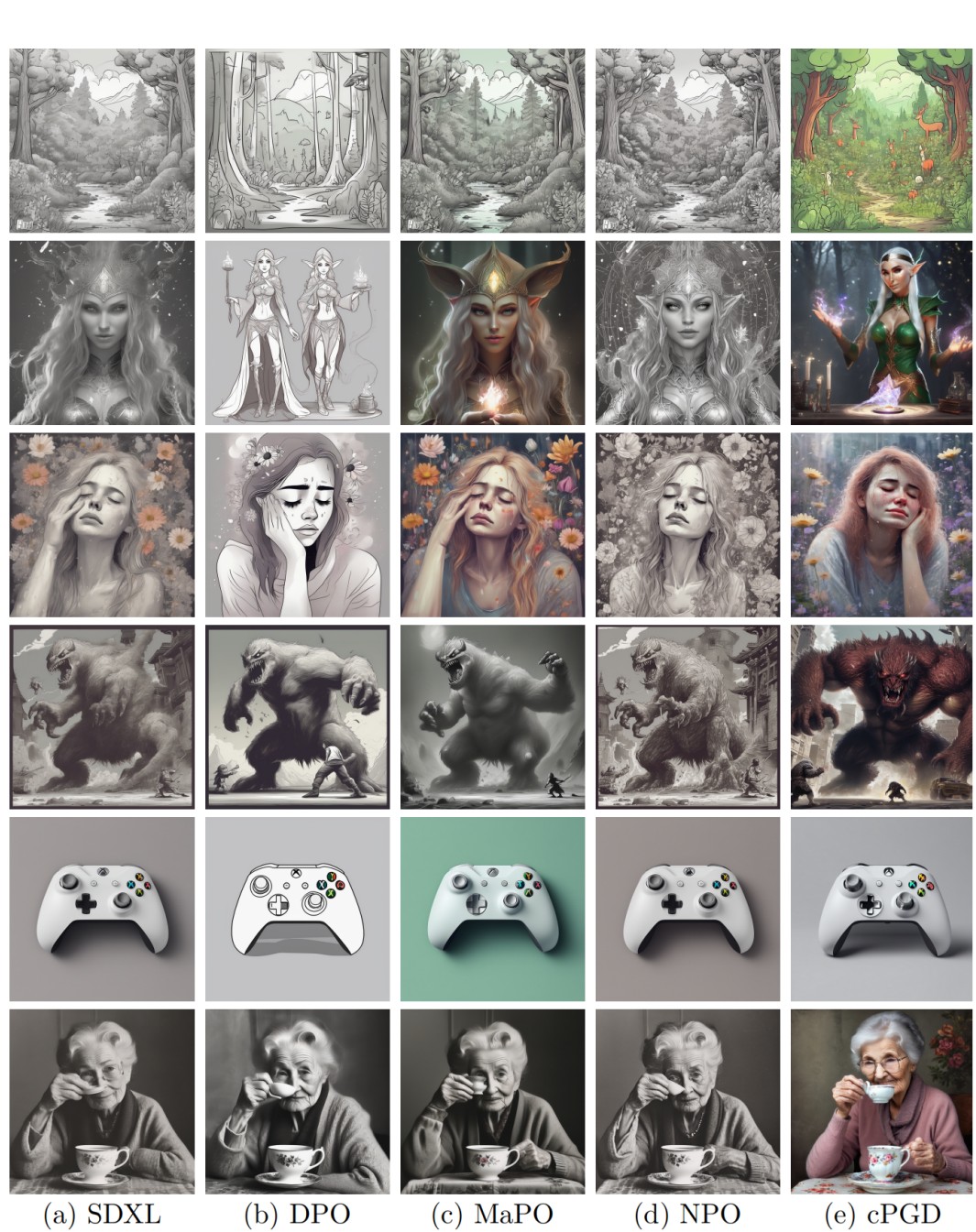

(a) SDXL  (b) DPO  (c) MaPO  (d) NPO  (e) cPGD

Figure 17: Comparison of preference-optimization methods on SDXL. Columns show outputs from the base model (SDXL), DPO, MaPO, NPO and cPGD.

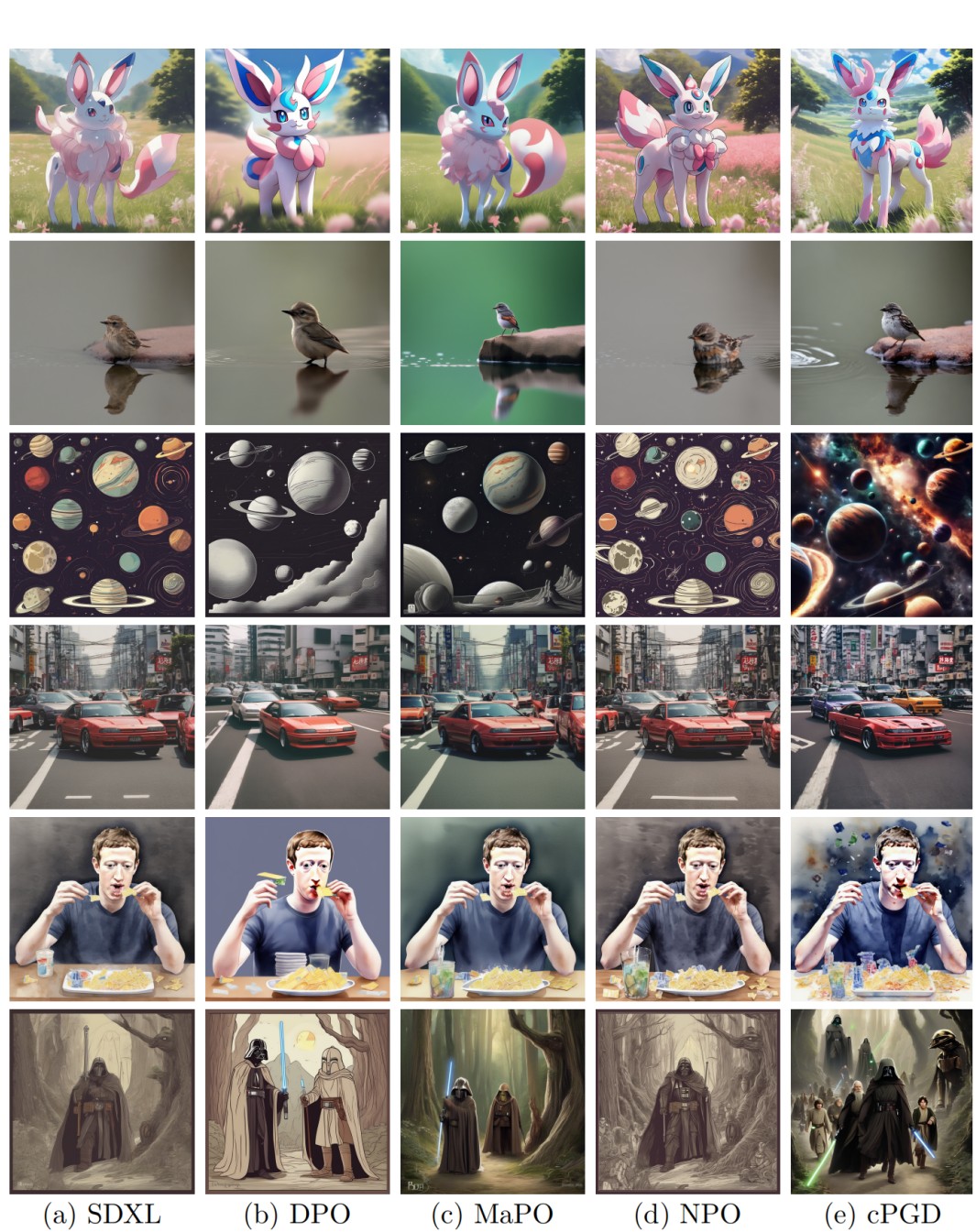

(a) SDXL     (b) DPO     (c) MaPO     (d) NPO     (e) cPGD

Figure 18: Comparison of preference-optimization methods on SDXL. Columns show outputs from the base model (SDXL), DPO, MaPO, NPO and cPGD.

