# OpenReview forum: "Rethinking Preference Alignment for Diffusion Models with Classifier-Free Guidance"
_ICLR.cc/2026/Conference — ICLR 2026 Conference Withdrawn Submission_

### Official Review · Reviewer_HMQp · 2025-10-30

**Soundness:** 2
**Presentation:** 2
**Contribution:** 2
**Rating:** 4
**Confidence:** 3

**Summary:**

This paper proposes a novel method that formulate preference alignment problem as classifier free guidance. Specifically, it proposes to use a preference-tuned policy model to provide guidance during inference. The author proposes two variants. The first variants PGD use the difference between the policy model prediction and a reference base model prediction as guidance signal. The second variant cPGD train two models separately on positive and negative samples of a preference dataset, and use the difference to provide guidance signal. Results show improvements across multiple metrics such as PickScore and CLIP metrics, using multiple base models including SDv1.5 and SDXL.

**Strengths:**

The method is straightforward and well-motivated. The presentation is clear. The experiments are comprehensive in terms of the metric covered, base model used, and baselines compared. The author also provided several ablation analyses on various design choices such as weighting.

**Weaknesses:**

There are three main concerns in terms of contribution and technical correctness.

First, the author assumes there are disjoint sets of positive samples and negative samples from a preference dataset, which is not true. For example, in Pick-a-Pick dataset, there are multiple images generated per prompt (say A,B,C,D), and there are preferences A > B and B > C, In such case, the positive and negative samples overlap with each other. This is fine for standard DPO method because its loss contrast **each preference pair** during the training. However, in cPGD, samples like B will appear in both positive and negative training set for cPGD. The author should provide detail discussions on how are these samples handled.

Second, some details of baseline evaluation are missing. For example, I find Figure 10 unconvincing since it reports that SDXL-DPO, which is trained on pick-a-pick, has a win rate <30% against base SDXL measured by PickScore, this greatly differs from the 89.4 win rate in Table 1. The author should provide more details on the distillation experiments. Since incorporating the proposed method will double the compute overhead, it is critical that the author provide careful analysis on the feasibility of the distillation approach.

**Questions:**

1. Why the "inference time is doubled"(L473)? In standard inference, we also need to predict cond_logits and uncond_logits as well. The only difference here is that we are running two models instead of one model. However, the total FLOPs should be the same? Is it possible to parallelize this inference process, provide both models can fit into GPU memory at the same time?

2. In cPGD, the author trained two separate models $\theta_{+}$, $\theta_{-}$ to provide guidance to the base policy $\pi_{ref}$, does this mean we actually need to run 3 models and the cost is tripled?

3. The cPGD approach is highly relevant to conditional SFT (cSFT) discussed in Diffusion-KTO literature. It seems the only difference is that in cSFT, a special conditional token "good" "bad" is added to the model during the training so that we can learn the distribution of both positive samples and negative samples in the same model, as opposed to use two models in cPGD. In general, it is also quite common to included negative images in the SFT training, with special tags like "deformed, low quality, etc". At inference, these tags are used as negative prompts in guidance. It seems that cPGD differs from these approaches only in that they have two separate models? The author should provide more thorough discussion on these well-established techniques and clarify what is the novel contribution of cPGD.

Overall, I find PGD a meaning contribution. The major concern is 1) the novelty of cPGD 2) the compute cost and the feasibility of distillation. If authors can address these points, I'm willing to increase my score.

---

> ### Author Response · Authors · 2025-11-14
>
> We appreciate the reviewer’s time and efforts in reviewing out paper and their acknowledegement on our contributions.
>
> Before answering the reviewer’s questions, we would like to point out an observation that previous work probably overlooks: DPO does not work well even on toy datasets. To illustrate this point, we conducted a simple toy experiment (Figure 1 in the revised draft): we construct the 8-Gaussians dataset in which 4 Gaussian distributions are labeled as positive and the rest 4 as negative. The preference pair dataset is therefore constructed by sampling random pairs in this 8-Gaussian dataset (for positive-postive and negative-negative pairs, we randomly sample a preference direction). We finetune a simple 3-layer-MLP base diffusion model with a batch size of 2048 with DPO and PGD. It is clearly shown that DPO is prone to overfitting and artifacts during the optimization process. PGD, in contrast, 1) can eliminate these artifacts due to the existance of the base model (in CFG) and 2) is more robust to overfitting because we may use large CFG scale to amplify the difference between finetuned and base models (thus avoiding finetuning for too long). We believe that this justifies both the intuition and novelty of our proposed method.
>
> > First, the author assumes there are disjoint sets of positive samples and negative samples from a preference dataset, which is not true. For example, in Pick-a-Pick dataset, there are multiple images generated per prompt (say A,B,C,D), and there are preferences A > B and B > C, In such case, the positive and negative samples overlap with each other. This is fine for standard DPO method because its loss contrast **each preference pair** during the training. However, in cPGD, samples like B will appear in both positive and negative training set for cPGD. The author should provide detail discussions on how are these samples handled.
>
> We agree that in pairwise datasets such as Pick-a-Pic, positive and negative samples naturally overlap (e.g., A > B and B > C makes B appear in both roles). However, **cPGD does not require disjoint sets**.
>
> Unlike Diffusion-KTO, which enforces strict separation via data cleaning, our method only assumes that the *average* quality of the positive set is higher than that of the negative set>an assumption that naturally holds in preference data. Therefore, we simply allow samples like “B” to appear in both set.
>
> Because both SFT models start from the same base and are trained with small learning rates, this overlap does not destabilize the contrastive guidance; empirically, we observe no degradation. We will clarify this relaxed assumption and explicitly note that **no additional data filtering is needed**.
>
> > Second, some details of baseline evaluation are missing. For example, I find Figure 10 unconvincing since it reports that SDXL-DPO, which is trained on pick-a-pick, has a win rate <30% against base SDXL measured by PickScore, this greatly differs from the 89.4 win rate in Table 1. The author should provide more details on the distillation experiments. Since incorporating the proposed method will double the compute overhead, it is critical that the author provide careful analysis on the feasibility of the distillation approach.
>
> We apologies for the misunderstanding here due to our inadequate descriptions. Indeed, **Figure 10 (now Figure 11) and Table 1 are evaluating different models and with different protocols, while Figure 10/11** evaluates **distilled student models** trained from the cPGD.  We have make this distinction explicit in the captions, and we have already **changed the bar chart to a pie chart** to better reflect that Figure 11 shows **proportions of win rate in three models**.

---

> > ### Author Response · Authors · 2025-11-14
> >
> > > Why the "inference time is doubled"(L473)? In standard inference, we also need to predict cond_logits and uncond_logits as well. The only difference here is that we are running two models instead of one model. However, the total FLOPs should be the same? Is it possible to parallelize this inference process, provide both models can fit into GPU memory at the same time?
> >
> > Indeed, for both prediction with text prompt and without text prompt (i.e., unconditional) we need to do this PGD operation (and this is partially why we use a different name instead of CFG to avoid confusion). This leads to doubled FLOPs.
> >
> > > In cPGD, the author trained two separate models , to provide guidance to the base policy , does this mean we actually need to run 3 models and the cost is tripled?
> >
> > To alleviate this issue, we conducted an experiment to distill a monolithic diffusion model out of our cPGD results and found (Fig.11 ) that empirically the distilled model is strictly preferred on all reward models and prompt datasets compared to the DPO baseline.
> >
> > > The cPGD approach is highly relevant to conditional SFT (cSFT) discussed in Diffusion-KTO literature. It seems the only difference is that in cSFT, a special conditional token "good" "bad" is added to the model during the training so that we can learn the distribution of both positive samples and negative samples in the same model, as opposed to use two models in cPGD. In general, it is also quite common to included negative images in the SFT training, with special tags like "deformed, low quality, etc". At inference, these tags are used as negative prompts in guidance. It seems that cPGD differs from these approaches only in that they have two separate models? The author should provide more thorough discussion on these well-established techniques and clarify what is the novel contribution of cPGD.
> >
> > We agree with the reviewer that cPGD is related to conditional-SFT approaches. Conceptually, however, cPGD relaxes the data assumptions: it does **not** require a clean, explicitly labeled split into “good” and “bad” samples, and naturally allows overlap between positive and negative roles (as discussed in Weakness 1). This makes cPGD less dependent on additional annotations or dataset cleaning.
> >
> > Moreover, empirically we observe that SFT-style methods (including cSFT) can be ineffective when the base model is already of higher quality than the preference data>for example, SDXL fine-tuned on Pick-a-Pic, where the base SDXL distribution often dominates the “positive” images. This phenomenon has also been reported in DSPO and MaPO, and cSFT does not fundamentally change the imitation-learning nature of SFT. In contrast, cPGD uses two light SFT branches only to estimate a contrastive score, and applies this at inference time on top of the strong base model, which is more robust in this regime.

---

> > > ### Author Response · Authors · 2025-11-14
> > >
> > > > Why the "inference time is doubled"(L473)? In standard inference, we also need to predict cond_logits and uncond_logits as well. The only difference here is that we are running two models instead of one model. However, the total FLOPs should be the same? Is it possible to parallelize this inference process, provide both models can fit into GPU memory at the same time?
> > >
> > > As a further clarification, the two branches can indeed be executed in parallel in the same way as standard CFG, provided both models fit into GPU memory. However, when the GPU is already running at high utilization (which is typical for large diffusion models), this parallelization does not actually reduce the effective inference time — it only changes how the workload is scheduled. Our statement about (roughly) doubled computational cost therefore refers to total FLOPs, rather than assuming any wall-clock speedup from parallel execution.

---

### Official Review · Reviewer_xwJS · 2025-10-30

**Soundness:** 3
**Presentation:** 2
**Contribution:** 1
**Rating:** 2
**Confidence:** 5

**Summary:**

The authors propose a simple approach for aligning diffusion models by training two separate models on positive and negative samples, respectively. During inference, these models are combined in a manner analogous to classifier-free guidance: at each denoising step, the method adds the positive model’s noise prediction, subtracts the negative one, and adds a reference model. The idea is clear, practical, and easy to implement.

**Strengths:**

1. The method is conceptually clear and intuitive — explicitly separating the contributions of positive and negative distributions provides a simple yet effective perspective on alignment.

2. The approach is easy to implement and has immediate practical value for production-level diffusion models.

3. The reported results are promising and demonstrate the potential of this straightforward modification.

**Weaknesses:**

1. Relying on two separate models is a major limitation. It is easy to imagine adding an additional conditioning to diffusion model and fine-tuning a single model instead, which would avoid the extra inference and memory overhead introduced by maintaining two networks.

2. Building on the previous point, the paper feels somewhat incomplete. It lacks deeper analysis or discussion of why the proposed method works. For example, does DPO actually fail to reduce the winner’s loss or increase the loser’s loss as intended? Is simple SFT on positive and negative samples truly optimal, or could training a standard DPO and a reverse DPO model for the positive and negative branches yield better separation and understanding?

**Questions:**

1. Could you provide an example of a single model trained in this manner, rather than maintaining two separate networks?

2. Could you elaborate on why the standard DPO framework underperforms compared to your proposed approach?

3. There may be more effective ways to model positive and negative distributions. For instance, have you considered training a standard DPO model for the positive samples and a reverse DPO model for the negatives (as an illustrative example)? If so, could you share any preliminary results or insights from such experiments?

I understand that these questions require additional experiments and are difficult to address under tight rebuttal deadlines. Nevertheless, I believe that exploring these points would significantly strengthen the paper and enhance its contribution to the community, particularly given its strong practical relevance. The authors have a clear opportunity to develop this idea into a true plug-and-play alternative to DPO that could gain broad adoption, but in its current form, the paper feels more like a strong hint of what it could become.

---

> ### Author Response · Authors · 2025-11-14
>
> We thank the reviewer for their time and efforts in reviewing out paper.
>
> > Relying on two separate models is a major limitation. It is easy to imagine adding an additional conditioning to diffusion model and fine-tuning a single model instead, which would avoid the extra inference and memory overhead introduced by maintaining two networks.
>
> > Could you provide an example of a single model trained in this manner, rather than maintaining two separate networks?
>
> We thank the reviewer pointing out this important issue and we agree that efficiency is essential. For this purpose, we conducted an experiment to distill a monolithic diffusion model out of our cPGD results and found (Fig. 11) that empirically the distilled model is strictly preferred on all reward models and prompt datasets compared to the DPO baseline.
>
> > Building on the previous point, the paper feels somewhat incomplete. It lacks deeper analysis or discussion of why the proposed method works. For example, does DPO actually fail to reduce the winner’s loss or increase the loser’s loss as intended? Is simple SFT on positive and negative samples truly optimal, or could training a standard DPO and a reverse DPO model for the positive and negative branches yield better separation and understanding?
>
> We thank the reviewer for brining up their concerns.
>
> To illustrate whether DPO behaves as intended, we present Figure 1 in the revised draft, which shows a 2D toy example where Diffusion-DPO can collapse, which we will include as part of the motivations for our proposed method.
>
> On whether simple SFT on positive/negative samples is reasonable, Figure 10 compares a **DPO-weighted SFT** variant (equivalent to a reparameterized DPO objective) with our **naive SFT** used in cPGD. Under identical settings, the simple SFT variant performs **better and more stably**. Section 4.2 further explains this via the reweighted DPO gradient view. We have made this design choice and its justification more explicit.
>
> Regarding “reverse DPO” branches: diffusion-NPO explores this idea and please see our reponse to the reviewer’s final question.
>
> > Could you elaborate on why the standard DPO framework underperforms compared to your proposed approach?
>
> To illustrate this point, we conducted a simple toy experiment (Figure 1 in the revised draft): we construct the 8-Gaussians dataset in which 4 Gaussian distributions are labeled as positive and the rest 4 as negative. The preference pair dataset is therefore constructed by sampling random pairs in this 8-Gaussian dataset (for positive-postive and negative-negative pairs, we randomly sample a preference direction). We finetune a simple 3-layer-MLP base diffusion model with a batch size of 2048 with DPO and PGD. It is clearly shown that DPO is prone to overfitting and artifacts during the optimization process. PGD, in contrast, 1) can eliminate these artifacts due to the existance of the base model (in CFG) and 2) is more robust to overfitting because we may use large CFG scale to amplify the difference between finetuned and base models (thus avoiding finetuning for too long).
>
> > There may be more effective ways to model positive and negative distributions. For instance, have you considered training a standard DPO model for the positive samples and a reverse DPO model for the negatives (as an illustrative example)? If so, could you share any preliminary results or insights from such experiments?
>
> Diffusion-NPO has explored this idea with“reverse DPO” branches, and we include related comparisons in Figure 5 and Table 1: a reverse-DPO-style model alone is not very competitive, but combined with the original DPO it achieves strong results (average win rate around 70%  with raw SDXL in Table 1). This indicates that richer positive/negative structures are promising, but due to time constraints we are not able to finish all experiments and present the results at this point.

---

### Official Review · Reviewer_7Zdk · 2025-10-31

**Soundness:** 3
**Presentation:** 3
**Contribution:** 2
**Rating:** 4
**Confidence:** 3

**Summary:**

This paper proposes Preference-Guided Diffusion (PGD) and Contrastive PGD (cPGD), which reformulate human preference alignment for text-to-image diffusion models as an inference-time guidance problem inspired by Classifier-Free Guidance (CFG). PGD combines the gradients of a base model and a DPO-finetuned model to achieve better preference alignment without additional training, while cPGD trains separate positive and negative branches and fuses them at inference for improved generalization. Experiments on Stable Diffusion 1.5 and SDXL show consistent Pareto improvements over Diffusion-DPO in reward, FID, and diversity, demonstrating a practical plug-and-play alignment framework.

**Strengths:**

1. This paper proposes two inference-time preference alignment methods inspired by the analogy to Classifier-Free Guidance (CFG), both conceptually simple yet effective.

2. PGD is a practical approach that can reuse existing pretrained and preference-finetuned weights.

3. The experimental results are extensive — across two prompt sets and multiple evaluation metrics, the effectiveness of the proposed methods is consistently demonstrated.

**Weaknesses:**

1. The proposed PGD formulation (Eq. 9) is merely defined by analogy to CFG, and the equation itself lacks theoretical justification.

2. The reason why overfitting is mitigated in cPGD is not experimentally verified (although a theoretical explanation is mentioned in Section 4.2).

3. Manual tuning of the guidance weight is required — it must be adjusted for each evaluation metric or dataset.

**Questions:**

1. Could you provide experimental evidence showing that cPGD indeed mitigates overfitting?
    For example, performance comparisons under different training data sizes could help demonstrate this effect.

2. Could you plot the influence of the guidance weight, as in Figure 6, for the other evaluation metrics as well?
    It would be helpful to see how consistent the curve shapes are across metrics.

3. The connection to NTK is a bit interesting. could you explain why PGD outperforms DPO under the case of slightly fine-tuned models?

---

> ### Author Response · Authors · 2025-11-14
>
> We thank the reviewer for their time and efforts in reviewing out paper.
>
> > The proposed PGD formulation (Eq. 9) is merely defined by analogy to CFG, and the equation itself lacks theoretical justification.
>
> The ideal objective of PGD is the same as that of DPO, basically to achieve $p(x) \propto p_\text{base}(x)R^{\beta}(x)$, and if we write take the nabla-log operation, turn things into the score function space and do some repameterization, we obtain the formulation of PGD with the CFG trick. We argue that it is the optimization and parametrization issues in large model finetuning that distinguish PGD from DPO.
>
> > The reason why overfitting is mitigated in cPGD is not experimentally verified (although a theoretical explanation is mentioned in Section 4.2)
>
> To illustrate this point, we show Figure 1 in the updated draft a toy experiment: we construct the 8-Gaussians dataset in which 4 Gaussian distributions are labeled as positive and the rest 4 as negative. The preference pair dataset is therefore constructed by sampling random pairs in this 8-Gaussian dataset (for positive-postive and negative-negative pairs, we randomly sample a preference direction). We finetune a simple 3-layer-MLP base diffusion model with a batch size of 2048 with DPO and PGD. It is clearly shown that DPO is prone to overfitting and artifacts during the optimization process. PGD, in contrast, 1)can eliminate these artifacts due to the existance of the base model (in CFG) and 2) is more robust to overfitting because we may use large CFG scale to amplify the difference between finetuned and base models (thus avoiding finetuning for too long).
>
> > Manual tuning of the guidance weight is required > it must be adjusted for each evaluation metric or dataset.
>
> Yes, our method does introduce a new hyperparameter, but we generally observe that optimal CFG scales are similar with respect to different reward models. Moreover, in practice (especially for artists that do DIY on open-source text-to-image models) CFG scale is commonly tuned. Our method does not introduce a huge overhead since it is just another CFG scale.
>
> > Could you provide experimental evidence showing that cPGD indeed mitigates overfitting? For example, performance comparisons under different training data sizes could help demonstrate this effect.
>
> We have studied behavior under different training data sizes (subset and fullset dataset) using HPDv3.  The exact split is detailed in Appendix A.3, and the quantitative results are reported in Table 3. Across both regimes, cPGD shows more stable gains over the base model and DPO-style baselines, while the DPO-style methods degrade more noticeably when trained on the smaller subset (a classical sign of overfitting). In contrast, cPGD retains its advantage in both the subset and full-set settings, which supports our claim that cPGD is more robust to limited or noisy preference data.
>
> > Could you plot the influence of the guidance weight, as in Figure 6, for the other evaluation metrics as well? It would be helpful to see how consistent the curve shapes are across metrics.
>
> Yes, we have visualized additional curves (Fig. 7 of the revised draft) and showed the dependence on guidance weight for all five preference metrics we report (PickScore, HPSv2, HPSv3, ImageReward, CLIP). These plots confirm that the curve shapes are strongly aligned across metrics.
>
> > The connection to NTK is a bit interesting. could you explain why PGD outperforms DPO under the case of slightly fine-tuned models?
>
> With PGD, one does not have to finetune the model too much because slight differences between the finetuned and base model are enough to tell which points in the base model distribution are more preferred, and we use the CFG scale to amplify it. This is essentially an unsupervised method: we only tune the “weights” on the data points, just that the weights come from lazy-regime finetuning > this is how PGD connects to NTK in a more intuitive way.

---

### Official Review · Reviewer_tVyu · 2025-11-03

**Soundness:** 2
**Presentation:** 3
**Contribution:** 2
**Rating:** 4
**Confidence:** 4

**Summary:**

This paper takes inspiration from CFG and proposes two methods, PGD and cPGD, for preference alignment. PGD treats the DPO-tuned model as the conditional model and the reference model as the unconditional model in CFG for inference. cPGD independently trains two models from positive and negative images separately and uses both models altogether in a CFG manner for inference. Experiments show that both methods improve preference-alignment performance compared with multiple baselines.

**Strengths:**

1. The idea of reinterpreting preference alignment in diffusion models through CFG is interesting. While CFG is typically used for conditioning on class labels or prompts, applying it to preference signals is both novel and practically useful. This reframing also enables plug-and-play alignment.
2. Presentation is clear, well-structured, and easy to follow.  The paper does a good job of walking the reader through the standard diffusion and DPO setups before introducing PGD and cPGD. Figures are informative and well-integrated with the text.
3. Extensive experiments showing that the proposed methods are effective and robust.

**Weaknesses:**

1. Even though the proposed PGD/cPGD are novel reinterpretations of preference alignment, the intuition is not well-explained. The paper lacks a formal verification of these methods. For example, what is the target distribution/posterior distribution that PGD/cPGD is trying to sample from? Can the authors justify that this interpolated guidance signal (between base and finetuned models) actually approximates the posterior over preferred samples?

2. The intuition behind cPGD is also underspecified. For example, why is it preferable to split the model into two, rather than learning a joint model with contrastive loss? Is there evidence that this separation improves the reward gradients or stabilizes the guidance signal?

As a result, although performance improves in practice, the mechanism behind that improvement is not fully explained. The methods appear somewhat ad hoc, and the absence of a more principled justification weakens the theoretical contribution.

**Questions:**

see Weaknesses

---

> ### Author Response · Authors · 2025-11-14
>
> We thank the reviewer for their time and efforts in reviewing out paper.
>
> > What is the target distribution/posterior distribution that PGD/cPGD is trying to sample from?
>
> The ideal objective of PGD is the same as that of DPO, since
>
> - The finetuned network is tuned with the DPO objective
> - Guidance with PGD essentially adjusts the $\beta$ parameter in $p(x) \propto p_\text{base}(x)R^{\beta}(x)$. Indeed if we write take the nabla-log operation, turn things into the score function space and do some repameterization, we obtain the formulation of PGD with the CFG trick.
>
> We further argue that it is the optimization and parametrization issues in large model finetuning that distinguish PGD from DPO, which can be seen in the toy experiment we added (Figure 1)
>
> > Why is it preferable to split the model into two, rather than learning a joint model with contrastive loss?
>
> To illustrate this point, we show Figure 1 in the updated draft a toy experiment: we construct the 8-Gaussians dataset in which 4 Gaussian distributions are labeled as positive and the rest 4 as negative. The preference pair dataset is therefore constructed by sampling random pairs in this 8-Gaussian dataset (for positive-postive and negative-negative pairs, we randomly sample a preference direction). We finetune a simple 3-layer-MLP base diffusion model with a batch size of 2048 with different methods. DPO results clearly show artifacts in the resulting distributions, while SFT distributions are far more clean. Therefore, composing SFT distributions is empirically better than DPO due to the highly nonlinear process of neural net finetuning.

---

### Author Response · Authors · 2025-11-14
**Paper withdrawal**

We decide to withdraw our paper. We sincerely thank the reviewer for investing time on our paper and providing valuable suggestions.

---

### Note · Authors · 2025-11-14

**Comment:**

We decide to withdraw our paper. We sincerely thank the reviewer for investing time on our paper and providing valuable suggestions.

**Withdrawal Confirmation:**

I have read and agree with the venue's withdrawal policy on behalf of myself and my co-authors.